# Neural Network for Aerosol Retrieval from Hyperspectral Imagery

Steffen Mauceri[1,2], Bruce Kindel[2], Steven Massie[2], Peter Pilewskie[1,2]

[1] Department of Atmospheric and Oceanic Sciences, CU Boulder, Boulder, Colorado, USA
[2] Laboratory for Atmospheric and Space Physics, Boulder, Colorado, USA

*Correspondence to*: Steffen Mauceri (Steffen.Mauceri@lasp.colorado.edu)

**Abstract.** We retrieve aerosol optical thickness (AOT) independently for brown carbon-, dust and sulfate from hyperspectral image data. The model, a neural network, is trained on atmospheric radiative transfer calculations from MODTRAN 6.0 with
10 varying aerosol- concentration and type, surface albedo, water vapor and viewing geometries. From a set of test radiative transfer calculations, we are able to retrieve AOT with a standard error of better than $\pm 0.05$. No a priori information of the surface albedo or atmospheric state is necessary for our model. We apply the model to AVIRIS-NG imagery from a recent campaign over India and demonstrate its performance under high and low aerosol loadings and different aerosol types.

## 1 Introduction

15 Remotely sensed surface spectral reflectance is used in many scientific disciplines including, geology, forestry, water studies and urban studies (Davis et al., 2002; Rencz and Ryerson, 1999). The surface reflectance can be either directly measured at the ground with portable field spectrometers or indirectly measured from air- and space-borne platforms. Observations at air- and space-borne instrument altitudes are sensitive not only to the signal from the surface but also the intervening atmosphere between surface and sensor. Thus, to derive surface reflectance from air- and space-borne observations, the data must be 20 corrected for atmospheric absorption and scattering effects. The main objective of atmospheric correction is the accurate removal of absorption and scattering by aerosols and gases. While absorption by water vapor and other gases is highly wavelength dependent, with relatively strong, discrete, absorption bands, aerosols extinction (the sum of absorption and scattering) is a smooth, continuous function of wavelength. This makes it challenging to separate it from the surface contribution. Most current atmospheric corrections ignore the aerosol variability within a scene. Instead, aerosol properties are 25 approximated from visibility (e.g., Gao, Heidebrecht and Goetz, 1993; Adler-Golden *et al.*, 1999) or derived from climatology. Such approximations can lead to large errors in retrieved surface reflectance, particularly for aerosol optical thickness (AOT) larger than 0.4, commonly found, for example, over east Asia (Bilal et al., 2014; Van Donkelaar et al., 2010). While instrument performance has steadily improved over the years, resulting in higher signal to noise ratios, improvements in the treatment of aerosols in atmospheric correction routines has not kept pace. To improve the retrieval of surface reflectance products from 30 air- or space-borne observations, the spatial variability of AOT and wavelength dependent single-scattering albedo and phase function or its moments within a scene have to be known.

Aerosols also pose a major uncertainty in climate predictions through their direct scattering and absorption of solar and thermal radiation and indirect effects on cloud albedo (Twomey, 1977) and clouds lifetime (Albrecht, 1989; Pincus and Baker, 1994). Their contribution to radiative forcing is now the biggest uncertainty to the total anthropogenic forcing between 1750 and 2011 (Myhre et al., 2013). The magnitude of the direct interaction of aerosol with radiation depends not only on their abundance,
but also their single scattering properties and the spectral reflectance of the underlying surface (Haywood and Boucher, 2000; Nan and A., 2015). Better quantification of the global distribution and optical properties of aerosols is a top priority to further improve climate projections.

Finally, aerosols are an important health risk factor (Pope III et al., 2009). For eastern Asia, the World Health Organization Air Quality $PM_{2.5}$ (amount of aerosols with a diameter less than 2.5 µm) Interim Target-1 (World Health Organization, 2006)
is exceeded for 50% of the population (Van Donkelaar et al., 2010), leading to an increase in mortality of approximately 15%. On a global scale, an estimated 7 million deaths were attributed to air pollution in 2016 (World Health Organisation, 2018). A better understanding of aerosol sources and their mixing in urban areas can inform decision makers and perhaps mitigate these hazards.

Currently, aerosols are routinely retrieved from ground and space-borne platforms. Ground based aerosol retrievals from *AErosol RObotic NETwork* (AERONET) (Holben et al., 1998) have the lowest uncertainty in retrieved AOT of less than 0.02 (Eck et al., 1999) but are spatially restricted. Space-borne instruments like the *Moderate resolution Imaging Spectroradiometer* (MODIS) (Salomonson et al., 1989) and the *Multiangle Imaging SpectroRadiometer* (MISR) (Diner et al., 1998) provide global coverage but retrievals from their measurements require separating the aerosol signal from the surface contribution. This results
in large differences between the derived aerosol products from different instruments (Chu et al., 2003; Levy et al., 2005, 2013; Prasad and Singh, 2007; Remer et al., 2005). Other approaches aim to use the vast information content from space-borne multiangle polarimetric observations that provide enhanced capability of separating aerosol signal from surface signal, and a better sensitivity to aerosol microphysical parameters. However, retrieving aerosol properties from such observations is highly complex and operational products have not yet reached the accuracy implied by theoretical calculations (Dubovik et al., 2019;
Kokhanovsky et al., 2015). Hence, accurate aerosol retrieval from space-borne platforms is still an active research topic.

To increase accuracy of global aerosol retrievals, we propose a retrieval algorithm that will be applicable to current and future hyperspectral space-borne instruments, such as Hyperspectral Precursor and Application Mission (PRISMA) (Labate et al., 2009), EO-1 Hyperion (Folkman et al., 2001), *Climate Absolute Radiance and Refractive Observatory* (CLARREO) (Wielicki
et al., 2013) and the *Hyperspectral Infrared Imager* (HyspIRI) (Lee et al., 2015). Exploiting the large data volumes and hundreds of spectral bands of these instruments requires new fast retrieval algorithms. To meet these needs, we propose using neural networks. In this study, we present a neural network that is used to independently retrieve dust, carbonaceous- and sulfate aerosols from hyperspectral imagery over land, with no a priori knowledge of the surface type or atmospheric state. The neural network can retrieve multiple collocated aerosol types and their contribution to the total AOT within a given scene.

After fitting the neural network parameters, also referred to as *training*, the model can be used to retrieve AOT in real time without further radiative transfer calculations. We apply the neural network to *Airborne Visible / Infrared Imaging Spectrometer Next Generation* (AVIRIS-NG) (Hamlin et al., 2011) imagery from a recent campaign over India and demonstrate its performance under high and low aerosol loadings and different aerosol types. AVIRIS-NG, a follow-on to the

*Airborne Visible / Infrared Imaging Spectrometer* (AVIRIS) (Green et al., 1998), has a spectral range of 380 – 2510 nm, a spectral resolution of 5 nm and spatial resolution of 4 m to 20 m depending on flight altitude.

The structure of our paper is as follows: Section 2 describes the forward radiative transfer calculations used to train an inverse model, the neural network. Sections 3 and 4 detail the architecture, training procedure and performance of the inverse model.

Furthermore, we explore how instrument noise and sampling resolution influence model performance. In Section 5 we apply the inverse model to AVIRIS-NG observations and compare results to AERONET and MODIS retrieved AOT. In section 6 we provide our conclusion.

**2 Forward Model**

To train a neural network for aerosol retrieval we need a dataset consisting of the model input – output pairs, or *samples*, of the inverse model. These samples need to span a wide variety of atmospheric states, viewing geometries and surface albedos. To generate such a dataset, we employ a forward model described in the following section.

**2.1 Radiative Transfer Calculations**

The forward model radiative transfer calculations were performed with the *MODerate spectral resolution atmospheric TRANSmittance algorithm and computer model* (MODTRAN) 6.0 (Berk et al., 2014) from 400 nm to 2500 nm. Multiple scattering was implemented with MODTRANs' DISORT algorithm (Stamnes et al., 1988), utilizing a conservative number of 32 streams. We chose the 'Tropical' atmospheric profile. The solar zenith angle (SZA) was varied between 25° and 50° and

water vapor varied between 0.4 g cm$^{-2}$ and 4.1 g cm$^{-2}$. The distance between ground and sensor (ground distance) was varied between 3 km and 6 km and the ground elevation between 0 m and 2000 m. The AOT at 550 nm varied between 0 and 1.0. Three types of aerosols, brown carbon, dust and sulfate (see Section 2.3) were modeled as an external mixture, with a fraction between 0 and 100%. For every parameter permutation we perform three radiative transfer calculations for a constant surface albedo of 0, 0.5 and 1. In the following, the calculated at sensor radiances for a surface albedo of 0, 0.5 and 1 are denoted as

$L_0$, $L_{0.5}$ and $L_1$. The three simulations are then used to calculate at sensor radiance for any given surface albedo utilizing the

MODTRAN interrogation technique (Verhoef and Bach, 2003). We first extract three atmospheric parameters, namely the spherical albedo, $\rho$, two-way transmittance, $\tau$, and path radiance, $L_P$:

$$\rho = \frac{2}{\frac{1}{f-1} + 2} \quad with \quad f = \frac{L_1 - L_0}{2 * (L_{0.5} - L_0)} \tag{1}$$

$$\tau = (L_1 - L_0) * (1 - \rho) \tag{2}$$

$$L_P = L_0 \tag{3}$$

Afterwards, we calculate the at sensor radiance, $L$, for the generated surface spectra, $r$ (see next Section):

$$L = L_P + \frac{\tau * r}{1 - r * \rho} \tag{4}$$

Finally, the radiance is convolved with a gaussian kernel with a full width half maximum (FWHM) of 5.6 nm in the UV and 5.8 nm in IR, similar to the AVIIRIS-NG spectral resolution.

## 2.2 Surface Spectra

To simulate a wide variety of surface types we need a multitude of surface spectra. However, the number of freely available surface spectra is limited. The risk of using too few surface spectra is that the model might not be able to extract general surface characteristics. Applied to scenes with a previously unseen surface spectra, the model would perform poorly. Furthermore, most catalogs provide pure surface spectra from pure surface materials, also referred to as endmember spectra. This case is not representative for most air- or space-borne observations over land where multiple surface types are present in a single instrument pixel. Therefore, we generate a catalog of mixed surface spectra by randomly combining a limited number of measured spectra from different sources. The combination is performed by taking the randomly weighted mean of two randomly chosen endmember spectra at a time, until we have a total of 100,000 mixed surface spectra.

Endmember spectra were obtained from https://ecosis.org/ and https://speclib.jpl.nasa.gov. The datasets include 844 vegetation reflectance spectra from Hawaii (Dennsion and Gardner, 2000), 173 vegetation spectra from Hawaii volcanoes national park

(Grimm, 2017), 1065 urban surfaces from Santa Barbara (Herold et al., 2004b) and 270 rock and soil spectra (Meerdink *et al.*, in prep.; Baldridge *et al.*, 2009). To remove high-frequency noise in the surface spectra due to low signal at some wavelengths (Herold et al., 2004a) we smooth the surface spectra with a Gaussian kernel as done by Thompson *et al.*, (2018).

An example of soil, sand and vegetation reflectances from the catalogs are shown on the left in Figure 1. The right side shows nine examples of how the three spectra are combined to generate mixed surface spectra.

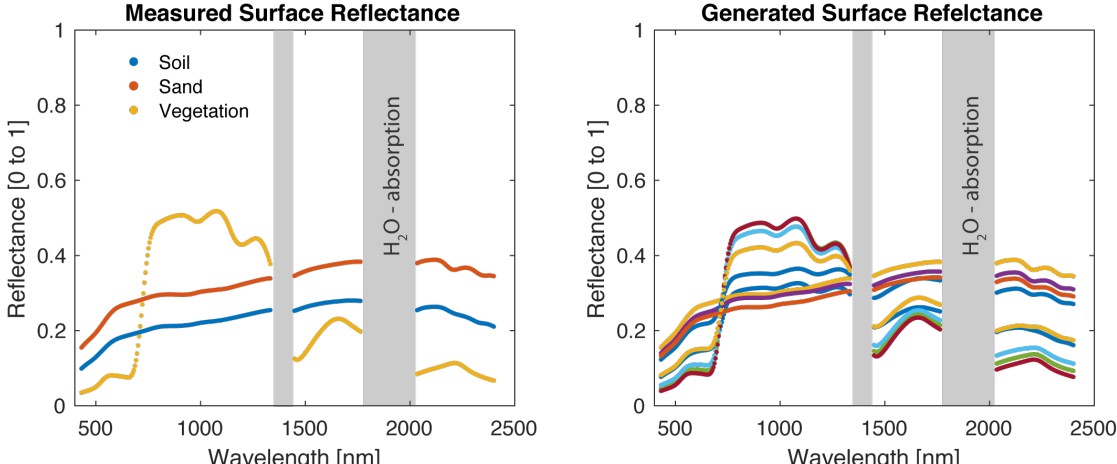

Figure 1: (Left): Surface reflectance for three different surface types (soil, sand and vegetation) from measured and smoothed surface spectra. (Right): Nine surface spectra, randomly generated from the three spectra on the left. Wavelengths with strong water vapor absorption are marked in grey.

## 2.3 Aerosol Parameterization

The optical properties of the three aerosols types that served as inputs to MODTRAN were calculated using three size distributions based upon Dubovik *et al.*, (2002) and the indices of refraction contained in HITRAN 2016 (Gordon et al., 2017). While aerosols cannot be strictly separated into types, we use these properties as representatives for dust, carbonaceous and sulfate aerosols. HITRAN 2016 includes $H_2SO_4$ indices at 300 K for sulfate aerosols and sand indices for dust aerosols from the AFCRL 1985 compilation (Fenn et al., 1985). The indices for sulfate and dust were selected because they cover the full wavelength range (0.2 to 40 μm) which is convenient for the use with MODTRAN. For brown carbon aerosols, from now on simply referred to as carbon, the indices are reported up to 1.2 μm by Alexander, Crozier and Anderson (2008). For longer wavelengths we extrapolated the real and imaginary parts.

Given the size distributions and the refractive indices, an extended version of the HITRAN-RI program (Massie and Hervig, 2013) was applied to calculate extinction, absorption, scattering spectra, and the Legendre moments of the phase function used in MODTRAN 6.0 calculations for carbon and sulfate. The calculations are based in Mie theory and thus assume homogenous

spherical aerosol particles. For dust we had to account for its non-spherical shape. We applied the T-matrix code of Mishchenko (Mishchenko and Travis, 1998), for randomly oriented particles, to generate the MODTRAN SAP files. The range of ratios of semi-major to semi-minor axes, or aspect ratios (AR), was varied between 1.01 and 1.8. This range contains the representative AR of 1.4 (Okada et al., 2001), while the aspect ratio of 1.01 corresponds to a nearly spherical particle. In our application of

the T-matrix code the second mode parameters (i.e. $Rad_2$=0.83 μm, $\sigma_2$=1.84, see Table 1) were used to specify the size distribution, and the AFCRL 1987 Sand indices are utilized.

Table 1 summarizes the inputs to the aerosol calculations, i.e. parameters for a size distribution with two log-normal distributions. Given the input size distribution and indices the resulting extinction spectra were distributed uniformly from the surface to 2 km altitude, with an additional stratospheric sulfate aerosol optical thickness of 0.006 distributed throughput the

stratosphere.

Table 1: Log-normal size distribution parameters for the three aerosol types considered

| Aerosol Type | $Den_1$ | $Rad_1$ | $\sigma_1$ | $Den_2$ | $Rad_2$ | $\sigma_2$ | Indices |
|---|---|---|---|---|---|---|---|
| Sulfate | 1.00 | 0.64 | 1.58 | 1.25 | 0.37 | 2.13 | AFCRL 1987 $H_2SO_4$ 300 K |
| Dust | - | - | - | - | 0.83 | 1.84 | AFCRL 1987 Sand |
| Brown Carbon | 1.00 | 0.086 | 1.49 | - | - | - | Alexander Brown Carbon |

To highlight the optical properties of the simulated aerosols, Figure 2 shows the MODTRAN simulated radiances for the three

different aerosol types overlying a black surface. The observed radiance is simulated at an altitude of 3 km with a SZA of 25° and ground elevation at sea level. An AOT of 1.0 was selected for each aerosol type. The single scattering albedos close to unity of sulfate and dust have a larger effect on the simulated radiance, compared to the lower single scattering albedo of carbon. For a highly reflective surface, the effects would be reversed, and we would see the strongest deviation from the case of no aerosols for carbon.

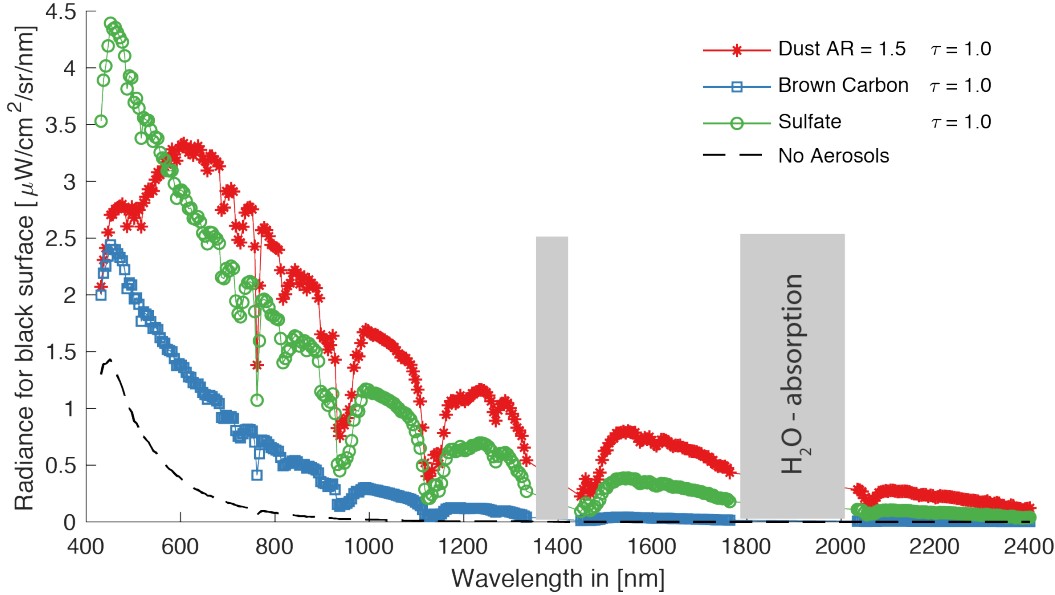

Figure 2: Radiance for a black surface, three different aerosol types and no aerosols, from an observed altitude of 3 km with a SZA of 25°. Every aerosol type (dust with an AR of 1.5, carbon and sulfate) is shown for an AOT of 1.0. No data are shown for the wavelengths of two water vapor absorption bands (grey bars).

## 2.4 Simulating Instrument Noise

Unlike radiative transfer calculations, the measured signal from real instruments contains noise. Therefore, we add noise to the radiative transfer calculations that is similar to the noise in the AVIRIS-NG instrument.

The noise is approximated by a three-parameter fit (see Equation 5) (Thompson et al., 2018) and derived from more complex

10    AVIRIR-NG noise models (Mouroulis et al., 2000, 2003; Tennant et al., 2008).

$$\sigma(L(\lambda), \lambda) \ = a(\lambda) * (b(\lambda) * L(\lambda))^{0.5} + c(\lambda) \tag{5}$$

The computed noise, $\sigma(L(\lambda), \lambda)$, is a function of three wavelength dependent parameters and observed radiance, $L(\lambda)$, which is a function of wavelength as well. Following a normal distribution, we randomly add the calculated noise to the radiative

15    transfer calculated radiances:

$$L_{noise}(\lambda) = L(\lambda) + \mathcal{X} \quad \text{with} \quad \mathcal{X} \sim \mathcal{N}(0, \sigma(L(\lambda), \lambda)) \tag{6}$$

On average, the signal to noise level for a typical scene is about 100 in the ultraviolet, 200 in the visible, 300 in the near infrared and peaks at about 1600 nm with a signal to noise level of 700.

## 3 Inverse Model

After using the forward model to generate a dataset to train the neural network, we are now able to train an inverse model that
relates radiance spectra to AOT for the three aerosol types.

### 3.1 Model Architecture

A subclass of neural networks, called *multilayer perceptrons*, have been shown to be able to approximate any linear or non-linear function (Hornik et al., 1989). After the training phase multilayer perceptrons can be used in real time at low
computational cost. This makes this model architecture ideal for our application. Neural networks have previously been used in many studies to extract information from remote sensing observations. For example to estimate cloud optical thickness and type (Minnis et al., 2016; Taravat et al., 2015), to un-mix surface types (Licciardi and Del Frate, 2011; Palsson et al., 2018) and to retrieve biophysical properties of vegetation (Verger et al., 2011; Xiao et al., 2014). Neural networks have also been applied to retrieve aerosol layer height from Ozone Monitoring Instrument (OMI) observations (Chimot et al., 2017), used to
estimate multiple aerosol parameters as the prior for an iterative Phillips-Tikhonov retrieval (Di Noia et al., 2017) and to estimate AOT from MODIS observations (Lary et al., 2009; Radosavljevic et al., 2010).

A multilayer perceptron is comprised of many individual operations, or *neurons*, that multiply their inputs by a matrix, or *weight*s, sum the results and add an additional vector, called *bias*. A non-linear function, the *activation function*, is applied to the results, or *output*s, of these neurons, permitting non-linear projections from the input space to the output space. In a network,
the output of neurons can be used as the input to other neurons. Hence, neurons are organized in *layers*. In general, more layers, and more neurons per layer, allow for more complex information retrieval. However, if the network becomes too complex for a given dataset and task, it will perform poorly for new model inputs. The right number of neurons and layers as well as other parameters, or *hyperparameters*, have to be determined empirically for every application. Thus, we altered the hyperparameters, trained the neural network on the majority of the samples, the *training set*, and evaluated the model
performance with samples that are separate from the training set, the *validation set*. Once we could no further reduce a user-defined *cost function*, we froze the hyperparameters.

Our neural network consists of five layers between the input and output layer, or *hidden* layers, containing 128 neurons in the first four hidden layers each and 96 neurons in the last hidden layer. The input layer consists of 322 neurons and the output
layer consists of 3 neurons. The first five layers are *fully connected*, meaning that all layer outputs are used as layer inputs of the succeeding layer. The sixth layer is separated into three groups with 32 neurons each (see Figure 3). The inputs to the

neural network are the radiance at 319 wavelengths, the SZA, ground distance and ground elevation. The output of the network is the independently retrieved AOT of the three aerosol types.

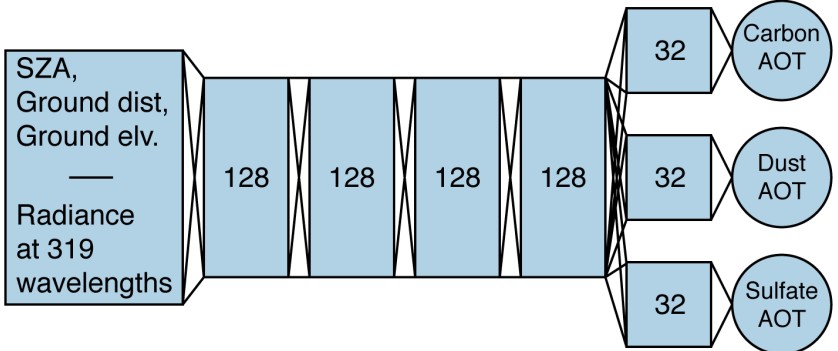

5        Figure 3: Model architecture of the neural network for aerosol retrieval. The inputs consist of the SZA, ground distance, ground elevation and the radiance at 319 individual wavelengths. The network has five hidden layers with 128 neurons in the first four layers and 96 neurons in the last hidden layer. The outputs of the network are the AOT for carbon-, dust- and sulfate aerosols.

10    To allow for non-linearities we add a rectified linear unit (ReLU) as the activation function, $g(x)$, which can be expressed as:

$$g(x) = \begin{cases} x, & if\ x \geq 0 \\ 0, & if\ x < 0 \end{cases} \tag{7}$$

where $x$ is the output of a neuron. During training we minimize the cost function, given by:

$$cost = \alpha * R(\theta) + \frac{1}{2n} \sum_{j=1}^{n} (\hat{Y}_j - Y_j)^2 \tag{8}$$

$$R(\theta) = \|\theta\|_2 = \sqrt{\sum_{i=1}^{m} \theta_i^2} \tag{9}$$

For our network $\hat{Y}_j$ and $Y_j$ are the $n$ true and predicted AOT, respectively. We further add the L2 norm $\|\theta\|_2$ to the vector of the $m$ neural network weights, $\theta$, to our cost function (see Equation 9), also referred to as *L2 regularization* or *weight decay*.

This helps to avoid overfitting to the training set. The L2 regularization term, $R(\theta)$, is weighted by $\alpha$ (see Equation 8) which is another hyperparameter that had to be determined empirically.

**3.2 Pre-processing**

From the 425 AVIRIS-NG channels we exclude calculated radiances at wavelengths with strong water vapor absorption. At these wavelengths, the majority of the surface spectra used in this study did not report or linear interpolate, surface reflectance. Furthermore, we exclude radiances at wavelengths that show strong signs of noise in the AVIRIS-NG data. A total of 319 wavelength channels remain. We then scale the radiance of a particular observation, $L_j$, by dividing through the cosine of the

SZA and multiplying with the square of the Sun-Earth distance, $d$, (Equation 10). This scales the magnitude of $L_j$ while preserving its spectral shape. Afterwards, we standardize the scaled observations, $R_j$, for the training process. During training, this results in a better conditioned cost function and allows the neural network to converge faster to a solution. Standardizing is performed by subtracting the mean, $\mu_\lambda$, and dividing by the standard deviation, $\sigma_\lambda$, at every wavelength, (Equation 11). The mean and standard deviation was calculated from the complete set of radiative transfer calculations.

$$R_j = \frac{L_j * d_j^2}{\cos(SZA)} \tag{10}$$

$$\overline{R_J} = \frac{R_j - \mu_\lambda}{\sigma_\lambda} \quad with \quad \mu_\lambda = mean(R_j)_\lambda \quad and \quad \sigma_\lambda = std(R_j)_\lambda \tag{11}$$

**3.3 Training, Validation and Test**

The MODTRAN radiance samples were split into a trainings, validation and test set. The validation and test set contain 10,000

randomly chosen samples each and the training set consists of 280,000 samples. Training is performed with Googles' TensorFlow framework (Abadi et al., 2016). We gradually minimize the cost function by adjusting the randomly initialized weights and bias terms with the gradient-based optimizer Adam from Kingma and Ba (2014), at a learning rate of 0.001. During training we evaluate the neural network performance on the validation set and update the model architecture and training parameters. Once, the cost function cannot be further minimized, training is complete.

## 4 Results and Discussion

After training of the neural network is completed, we evaluate its performance on the test set. For the samples in the test set, that were not present during training, we find a linear correlation coefficient of 0.87, 0.98 and 0.96 for the AOT of carbon, dust and sulfate, respectively (see Figure 4). The standard error for carbon-, dust- and sulfate aerosols is 0.05, 0.02 and 0.03, respectively. Thus, the model accuracy is higher for dust and sulfate, which have a larger single scattering albedo compared to carbon.

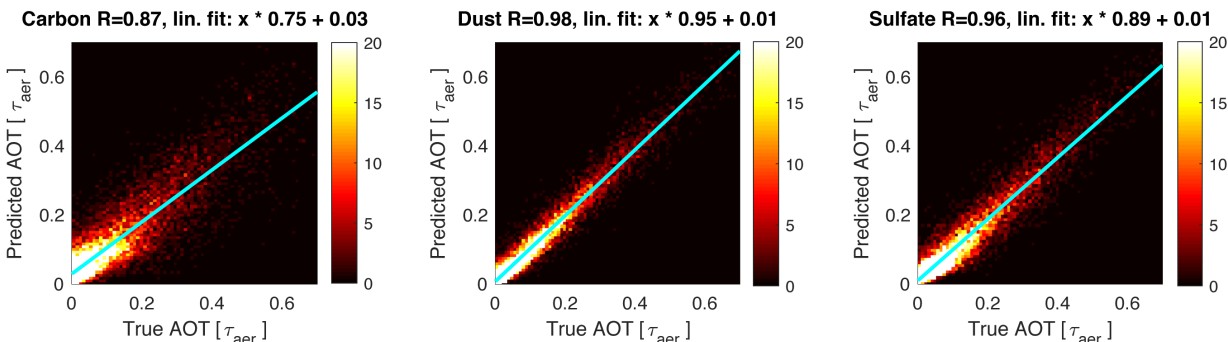

Figure 4: AOT for carbon, dust and sulfate aerosols, retrieved by the model vs true AOT from the test set. The cyan line shows the linear fit to the data with slope and y-intercept given in the respective titles.

We further investigate the model's performance for retrieved AOT under varying amounts of the three aerosol types. The absolute error in retrieved AOT for the three aerosol types is shown in the top row of Figure 5. Horizontal gradients (vertical bands) indicate that the model's performance for the retrieval of a single aerosol type depends on the concentrations of the other aerosols in a given observation. Vertical gradients indicate that the model's performance is dependent on the AOT of the aerosol that we are trying to retrieve. For the error on retrieved carbon (Figure 5 a) and sulfate (Figure 5 c) we find dependencies on AOT while the error in the retrieval for dust (Figure 5 b) appears insensitive to its AOT. Examining the retrieval error in percent of AOT (bottom row) we find that all three aerosol retrievals have higher relative errors for lower AOT and a standard error of about 40% for an AOT of 0.1. We further analyzed the model's performance over a SZA range from 25° – 50°, ground elevation from 0 – 2000 m and ground distance from 3000 – 6000 m. No significant correlation between model error and the three parameters was found.

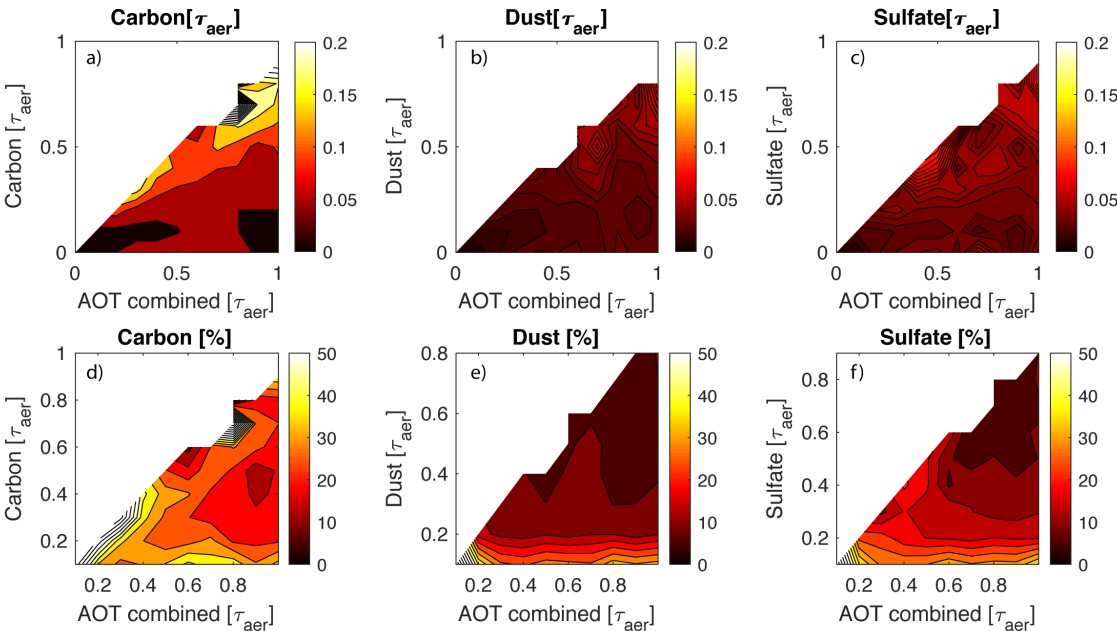

Figure 5: Error in retrieved AOT for carbon, dust and sulfate aerosols on the test set. (a, b, c) shows the absolute error while (d, e, f) shows the error in [%]. The color-mapping is held constant for each row and varies across the three columns.

## 4.1 Model Performance for Varying Surface Types

To investigate systematic, surface dependent biases in the model we derive AOT for the three aerosol types over various unmixed surface types. The data consist of 250,000 samples. The standard error and mean between true and predicted AOT for different surfaces types is summarized in Figure 6. For the retrieval of carbon, we find the largest standard error for asphalt

10  with $\pm0.08$ and the largest systematic bias for grass of $+0.02$. For dust the largest systematic bias is less than $+0.01$ and occurs for scenes with concrete. The standard error is similar for all surface types and approximately $\pm0.02$. The systematic biases for the retrieval of sulfate aerosols are mostly negative with asphalt and concrete causing the largest bias of $-0.01$. Overall, the standard error for the retrieval of carbon over most surfaces is larger compared to the other two aerosol types. This is not surprising, considering the overall lower performance of the model for the retrieval of carbon aerosols. Note that the model's

15  performance should be evaluated from the more realistic case of mixed surface spectra as was done in the previous section.

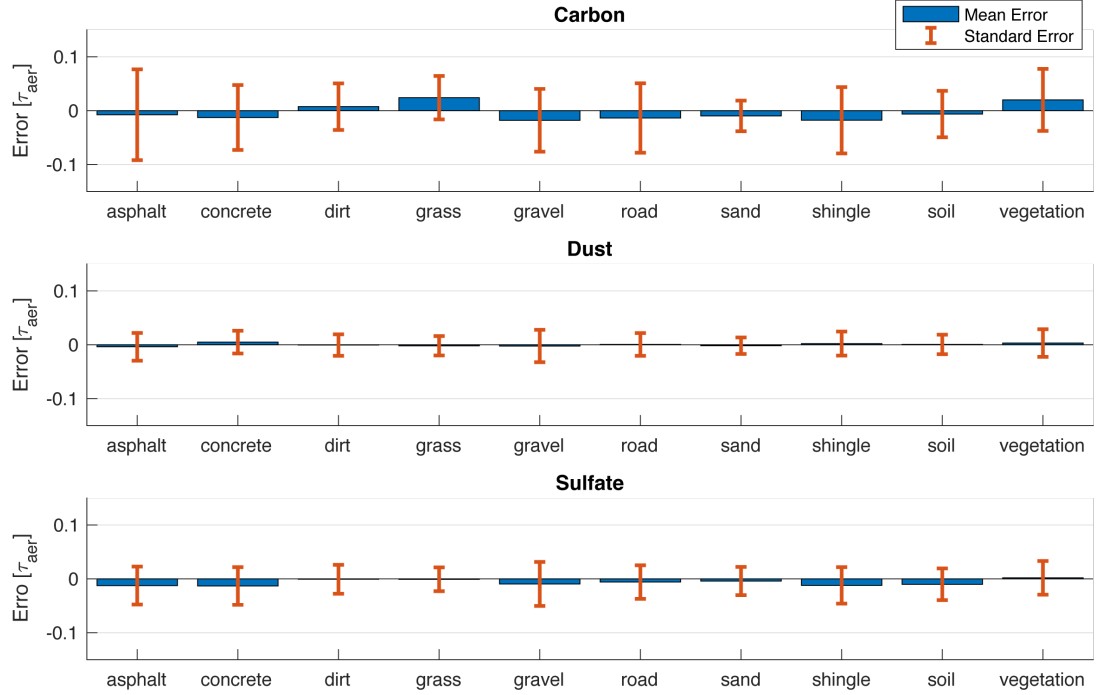

Figure 6: Mean error and standard error of retrieved AOT for carbon, dust and sulfate for various unmixed surface types from the test set.

## 4.2 Effect of Spectral Resolution, Sampling Resolution and Instrument Noise

Here we examine how spectral resolution, sampling resolution and instrument noise affect model performance. The underlying motivation is to estimate the model's performance for instruments other than AVIRIS-NG, which might have a 10 nm spectral resolution, fewer wavelength channels as well as a higher or lower signal to noise ratio. Hence, we train and analyze the model's performance for an additional 23 networks with varying noise, spectral resolution and sampling resolution.

To simulate the fewer wavelength bands, the training-samples were reduced in sampling resolution, leaving 319, 107, 36 and 12 uniformly spaced, wavelengths per sample. Furthermore, to account for different signal to noise ratios, we changed the simulated AVIRIS-NG equivalent noise level (see Equation 5 and 6) by multiplying it with 0 (no noise), 1 and 3 before applying it to our training and test samples. Finally, we preformed all calculations once for the AVIRIS-NG spectral resolution of approximately 5 nm and for a spectral resolution of 10 nm. All neural network parameters were kept constant, except the input layer, which had to be adapted to the reduced number of wavelengths. Training was stopped when the error on the validation set could not be reduced any further or we reached a maximum of 10,000 epochs, meaning that every training-sample was used during training 10,000 times. While we found dependencies of retrieval performance to varying amounts of

noise and number of wavelength channels, the spectral resolution had no significant effect. On average the models trained with a spectral resolution of 5 nm had a standard error in retrieved AOT that was only 0.001 smaller than for the cases with a spectral resolution of 10 nm. Therefore, we limit the following discussion to the results of the 12 neural networks trained on radiative transfer calculations with the AVIRIS-NG spectral resolution of approximately 5 nm and note that these values are

also representative for an instrument with a 10 nm spectral resolution.

The standard error on the test set of the respective 12 neural networks is shown in Figure 7. The left column shows the standard error for the complete test set (AOT is varied between 0 and 1) while the right column shows the standard error for low aerosol loadings, with AOT ranging between 0 and 0.3. As expected, we find a decrease in model accuracy for fewer wavelengths and more noise. This decrease in model accuracy, with respect to the idealized case of 319 wavelength bands and no noise, is

nearly symmetrical for our chosen test cases. Thus, if we reduce the number of wavelength bands by a factor of three the model has  similar accuracy compared to if we add AVIRIS-NG equivalent noise and if we reduce the number of wavelength bands by a factor of nine the model has similar accuracy compared to applying three times AVIRIS-NG equivalent noise, and so on. This holds true for all aerosol types. Overall, the model has the highest accuracy for the retrieval of dust. To put the calculated standard errors in the left column into perspective: if the model would randomly guess the combined AOT of all three aerosols

between 0 and 1 and simply divide by three, the standard error would be $\pm 0.10$. Thus, all trained models show higher accuracy than guessing randomly. If we had a model that would be able to retrieve the combined AOT without error, and then simply divide by three, the standard error would be $\pm 0.07$. For the retrieval of carbon, the models with 12 wavelengths bands and 3 times AVIRIS-NG equivalent noise shows such a standard error. This is an indication that the AOT from carbon aerosols cannot be isolated from other aerosols for instruments with only 12 wavelengths and 3 times AVIRIS-NG equivalent noise.

The retrieval of dust and sulfate requires fewer wavelength bands and can tolerate more noise compared to the retrieval of carbon.

For aerosol retrieval under low AOT conditions (right column in Figure 7), a model that would guess the combined AOT randomly between 0 to 0.3 and divide by three, would have a standard error of $\pm 0.06$ and a model that can determine the combined AOT perfectly and then simply divides by three would have a standard error of $\pm 0.03$. Most combinations of

wavelength bands and instrument noise have standard errors that exceeds this threshold of $\pm 0.03$ for the retrieval of carbon. This highlights the limitations of the model for the separation of carbon aerosols for low levels of AOT. Additionally, it stresses the importance of low noise hyperspectral instruments, such as AVIRIS-NG.

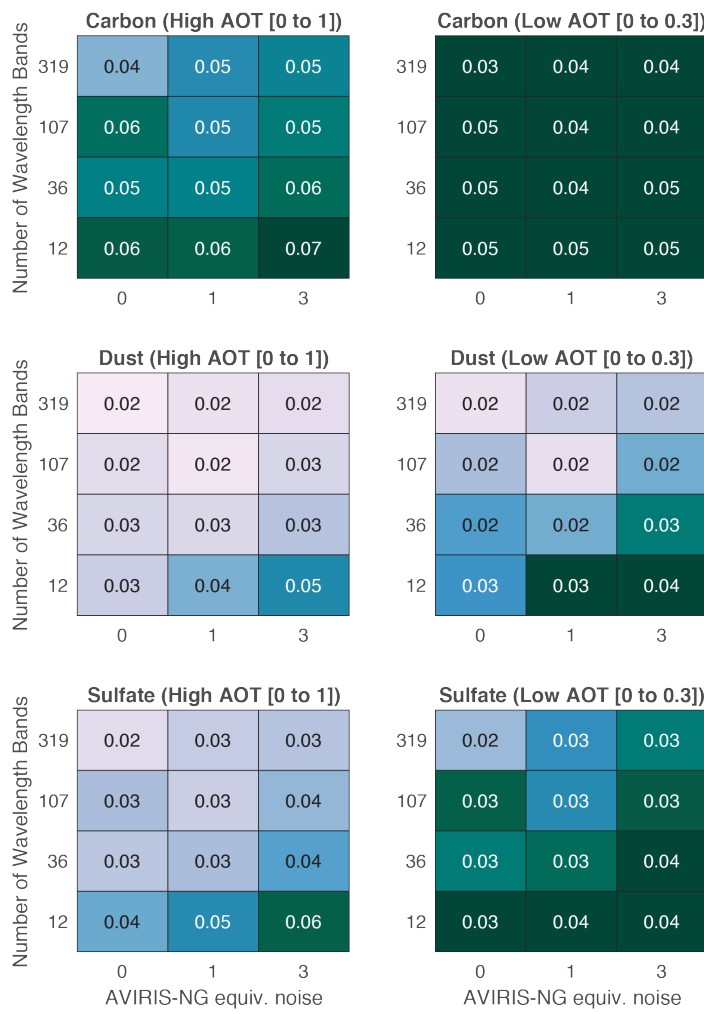

Figure 7: Standard error for retrieved AOT of 12 individually trained neural networks with (319, 107, 36, 12) wavelength bands and varying amount of simulated AVIRIS-NG equivalent noise (0, 1, 3) (see Equation 5 and 6) from the test set. Left column shows the standard error when AOT is varied between 0 and 1. Right column shows the standard error for AOT between 0 and 0.3.

### 4.3 Sensitivity Analysis

It is inherently difficult to interpret the inner workings of neural networks. However, by perturbing the inputs and observing the changes of the outputs one can infer the relative importance of an input for a given model (Blackwell, 2012). We perform such a sensitivity analysis by increasing one input at a time by 1%, while keeping the other 321 inputs unchanged. The model output is then calculated for the entire test set and compared to the retrieval without the perturbation. For example, AOT of carbon is derived while the model input, representing the observed radiance at 500 nm, is increased by 1%. All other model

inputs, for example radiance at 600 nm and 700 nm or SZA, are kept unchanged. We perform such a sensitivity analysis once for the model trained without noise (an ideal instrument) and once for the model trained with AVIRIS-NG equivalent noise. The sensitivity to every input is shown in Figure 8. For the model trained without noise (top, third and fifth row) we find more sensitivity at 687 nm and 762 – 767 nm for the retrieval of carbon and dust while sulfate shows more sensitivity to the latter.

These wavelengths correspond to the oxygen B-and A-band located at 685 nm – 695 nm and 759 - 771 nm, respectively. Multiple studies have suggested the use of these absorption bands for the retrieval of AOT and its vertical structure (Dubuisson et al., 2009; Heidinger and Stephens, 2000; Min et al., 2004). The sensitivity to small perturbations of SZA, ground distance and ground elevation is small compared to the radiances. From these three model inputs, surface elevation is indicated to be the most important for the retrieval of dust and sulfate.

For the model trained with AVIRIS-NG equivalent noise we find approximately an order of magnitude lower sensitivity at shorter wavelengths compared to their respective counterparts trained without noise (Note the different y-scales for the six sensitivity plots). This demonstrates how the model adapted to small perturbations (noise) at individual wavelengths by becoming less sensitive to these perturbations. For longer wavelengths, the change in sensitivity is less pronounced. In general, we observe a relative shift in sensitivity from shorter towards longer wavelengths when instrument noise is added. The shift

in sensitivity to longer wavelengths might be a direct effect of the noise distribution of AVIRIS-NG which allows for a higher signal to noise ratio at longer wavelengths. Additionally, there is an overall smoother shift in sensitivity between neighboring wavelengths. This can be interpreted as the model relying on multiple neighboring wavelengths to obtain their shared information content, rather than interpreting wavelengths individually.

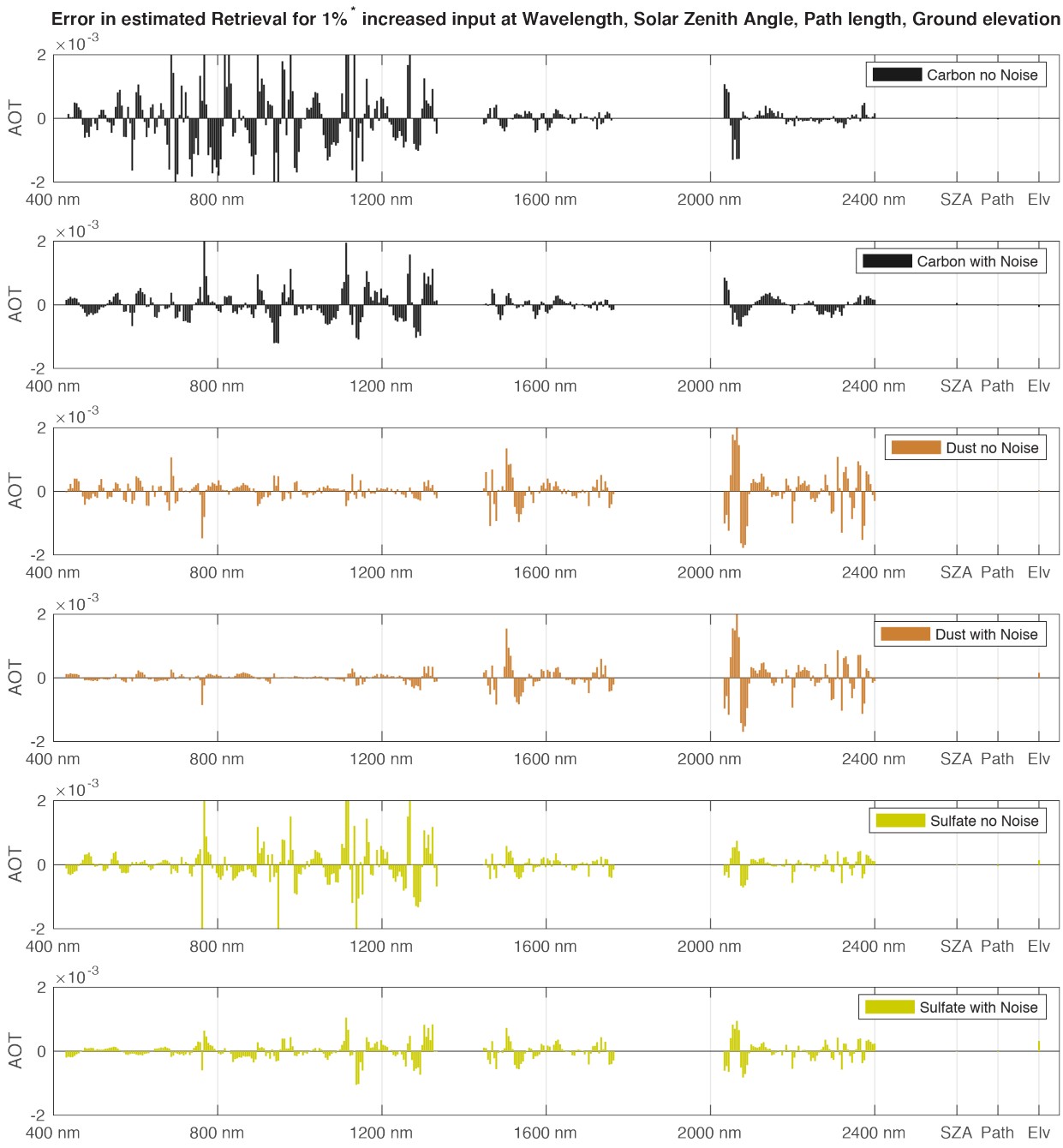

Figure 8: Sensitivity for retrieved AOT of carbon, dust and sulfate to all model inputs. The x-axis shows the model inputs (radiances at shown wavelength, SZA, ground distance and ground elevation). The y-axis shows the difference in retrieved AOT when increasing a given input by 1% while keeping all other inputs unchanged.

## 5 Applying the Model to Real Imagery

We apply the trained neural network to AVIRIS-NG observations from a flight campaign in 2016 over India in collaboration with Space Applications Centre, Indian Space Research Organization (SAC, ISRO). The results are compared to MODIS and AERONET retrieved AOT and a reanalysis product.

### 5.1 Preprocessing of AVIRIS-NG Observations

To remove remaining noise in the AVIRIS-NG observations we use a principal component analysis (PCA) (Wold et al., 1987) and inspect the generated eigen-images manually. The PCA is only applied to the 319 wavelength channels that we used to train the model on. As stated before, these channels were down selected from the 425 AVIRIS-NG channels to avoid wavelength bands with strong water absorption and instrument noise. The first 16 components explain approximately 99.9% of the variability and are dominated by image features (see Figure 9). Most higher principal components are dominated by systematic noise (vertical stripes along the flight path). We reconstruct the AVIRIS-NG observed radiances from these first 16 principal components. This effectively removes principal components higher than 16 from all analyzed AVIRIS-NG imagery. Afterwards, the radiance for every pixel is treated as an independent observation and scaled and standardized (Equation 10 and 11) to match the training set. We acknowledge that the choice of retaining the first 16 principal components is rather arbitrary and should ideally be made on a per flight basis. However, for practical reasons we decided to use one threshold for all imagery considered in this study. The threshold is a tradeoff between removing valuable information and reducing noise. Experiments with more and fewer principal components indicated that the model was insensitive to the exact number of remaining principal components.

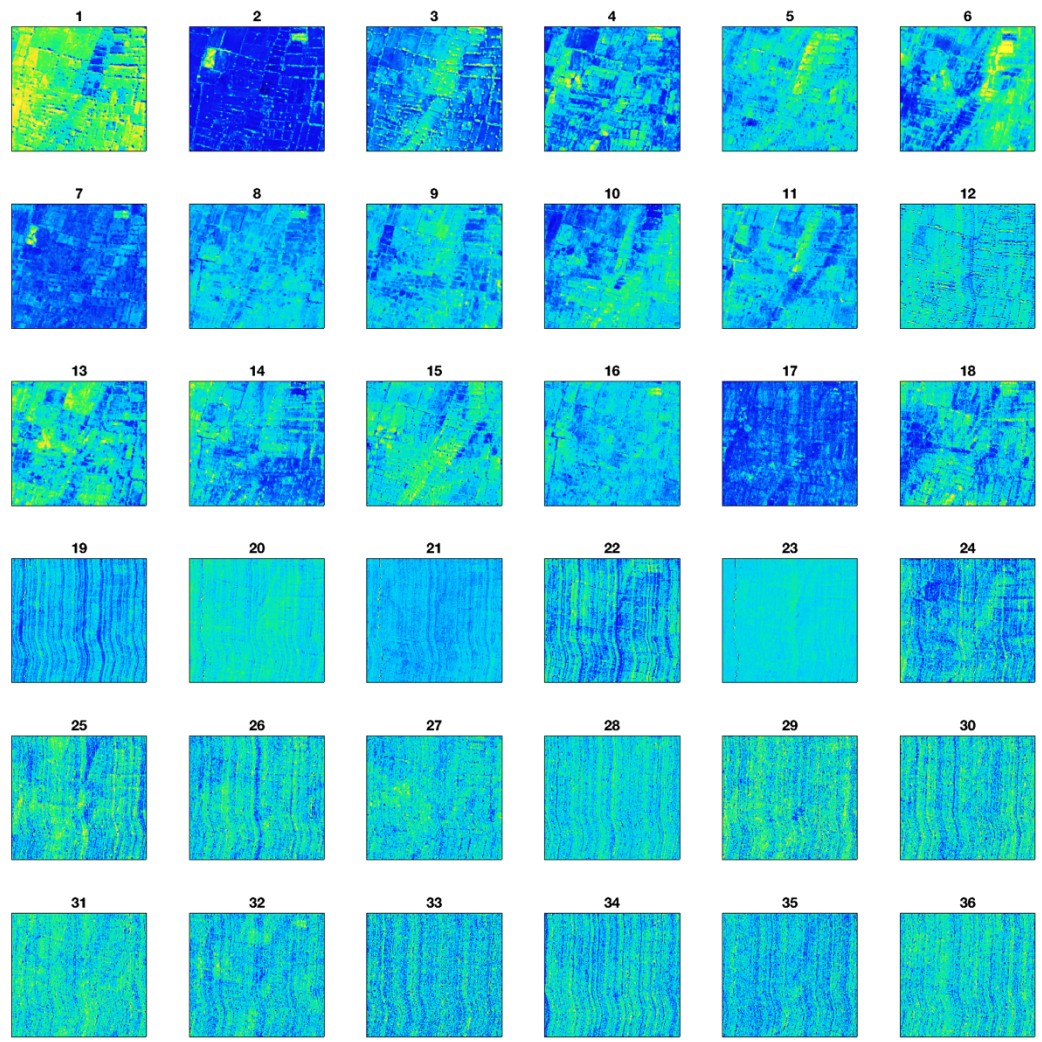

Figure 9: First 36 eigen-images from an AVIRIS-NG flight on 10/01/2016 near Coimbatore, India. Shown is a spatially resolved scene of 100 x 100 ground pixel, approximately 500 x 500 m. Instrument artifacts (vertical stripes) are visible for eigen-images greater than 16 (for example 19 and 22).

## 5.2 Novelty Detection

Our model is trained on a limited set of training examples. The set of surface types available for training is not complete. Generally speaking, library spectra of surface materials vastly under-represent the spectral variability of surface materials found in nature. The variety of surface materials is just too great to include in any single library. Applying the model to scenes with new surface types, which have significant differences compared to the surface types in the training set can lead to false

aerosol retrieval by the model. Hence, it is important to measure the similarity of a given AVIRIS-NG scene to the training examples and discard individual image pixel that are far outside of the training space. This is referred to as *novelty detection*. For this purpose, we train a second neural network proposed by Japkowicz et al., (1995) on the training samples with AVIRIS-NG equivalent noise. The network architecture is an auto-associative multilayer perceptron (Kramer, 1992) with three hidden

layers and shown in Figure 10. All three hidden layers use a ReLU activation function and consist of 512, 32 and 512 neurons, each. The input- and output-layer consist of 322 neurons, each. The network takes 319 radiances at individual wavelengths (measurements of one image pixel), SZA, ground distance and ground elevation as input parameters and is trained to reproduce these parameters after some computation by the network. The network is trained in a manner similar to the model for aerosol retrieval and uses the same optimization algorithm and cost function (see Equation 8) with $n = 322$, and $\hat{Y}_j$ and $Y_j$ being the

original and reproduced radiances and SZA, ground distance and ground elevation. The first three layers (Input, Compression and Bottleneck) act similarly to deriving the first 32 principal components but are non-linear. The last two layers (Decompression and Output) can be interpreted as reproducing the radiances only from their first 32 principal components, but again, are non-linear. After the replication of the input parameters we compare those to the original inputs and calculate the mean square error between the two. During training, the neural network learns to minimize this error. For example, the

neural network learns that the radiance at 2100 nm is highly correlated with the radiance at 2300 nm. Thus, it can reconstruct (decompress) both radiances with only one value passed in from the Bottleneck layer with little error. Once the neural network is trained and applied to previously unseen features it will compress and decompress features that are similar to the training set (high correlation between 2100 nm and 2300 nm) with a smaller error than features that are different (low correlation between 2100 nm and 2300 nm). Finally, a threshold for the error is determined as a tradeoff between the number of remaining

aerosol retrieval and the number of remaining outliers. Samples above the determined threshold are considered new and not considered for the aerosol retrieval.

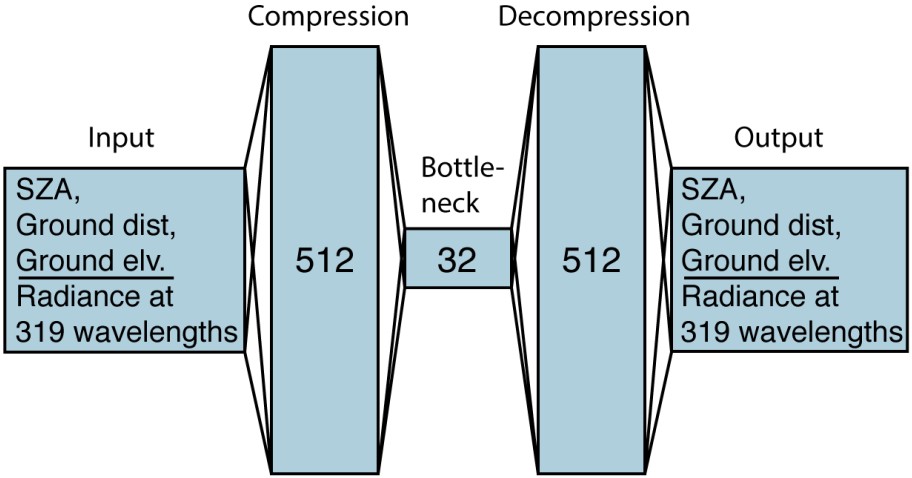

Figure 10: Auto-associative neural network for novelty detection used for novelty detection. The input and output layer consist of SZA, ground distance, ground elevation and radiances at 319 wavelengths. The network has three hidden layers with 512, 32, 512 neurons per layer.

## 5.3 Results

Figure 11 and Figure 12 show the aerosol retrieval for two of the 21 analyzed AVIRIS-NG scenes. The scene in Figure 11 was captured on 02/04/2016 near Kota, India. It shows a detail of the flight with 100 x 500 pixels and an approximate ground resolution of 5 m per pixel. The median and standard deviation of the retrieval is indicated at the top of the first four panels, showing the combined AOT and un-mixed AOT for carbon, dust and sulfate. The normalized mean square error from the auto-associative neural network for novelty detection and a true color image is shown on the right as well. Image pixels that lie above a user defined threshold are highlighted in red and discarded. For the scene shown in Figure 11 the discarded image pixels consist of water features in the middle and bottom portions of the scene as well as some agricultural sites. The detection of water by the neural network for novelty detection is to be expected, since the spectral shape of water is very different to most land surfaces and was not part of the training set. The aerosol retrieval still includes surface features. For example, it overestimates carbon aerosols over what appears to be a street (middle of second plot from the left). Some residual surface features are not entirely unexpected as less challenging atmospheric retrievals from imaging spectroscopy, for example water vapor (Thompson et al., 2015), often contain surface reflectance artifacts. The detail shown in Figure 12 is from an AVIRIS-NG flight near Gundlupet, India from 01/10/2016. The model for novelty detection excluded mostly individual fields with bare soil. Similar to the figure above, we find some residual surface features in the retrieval. Both images show the limitation of the model in distinguishing small variations in AOT from different surface types. To minimize the residual surface features a median filter could be applied in post processing at the cost of lower spatial resolution.

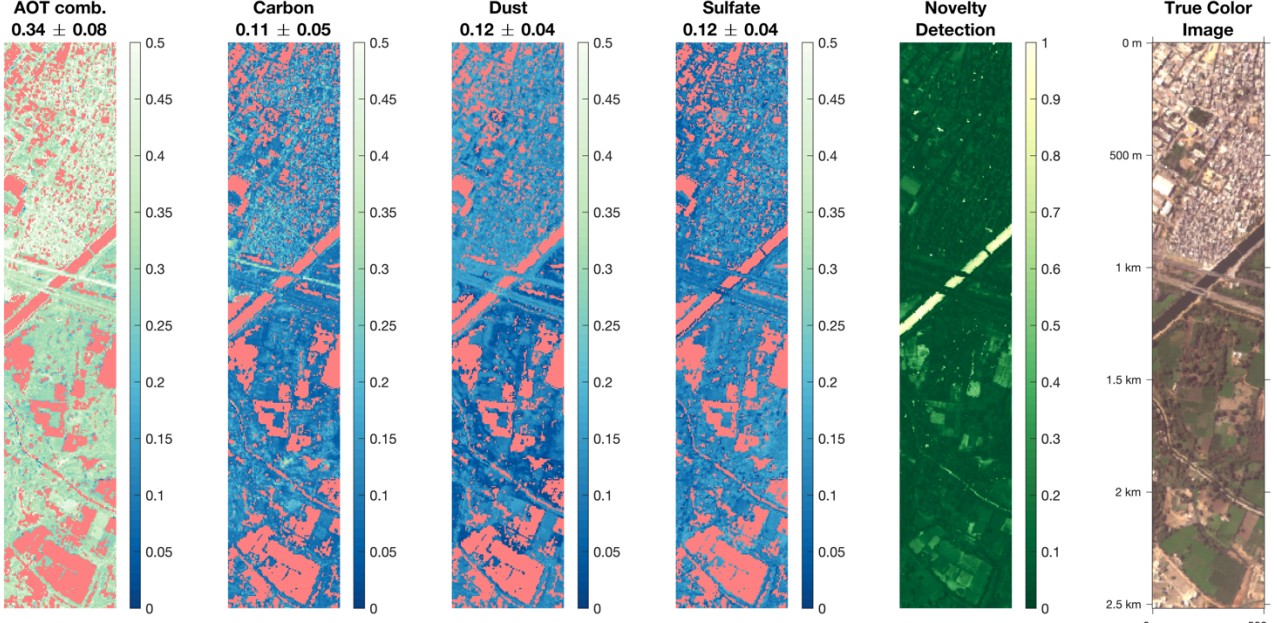

Figure 11: Aerosol retrieval with the model from AVIRIS-NG imagery near Kota, India, 02/04/2016. The median and standard deviation of the retrieval is indicated at the top of each panel. The normalized output of the neural network for novelty detection is shown in the panel, second from the right. Values above a chosen threshold are discarded from the aerosol retrievals and highlighted in red (e.g. a river in the middle of the images). A true color image of the scene is shown as well for reference.

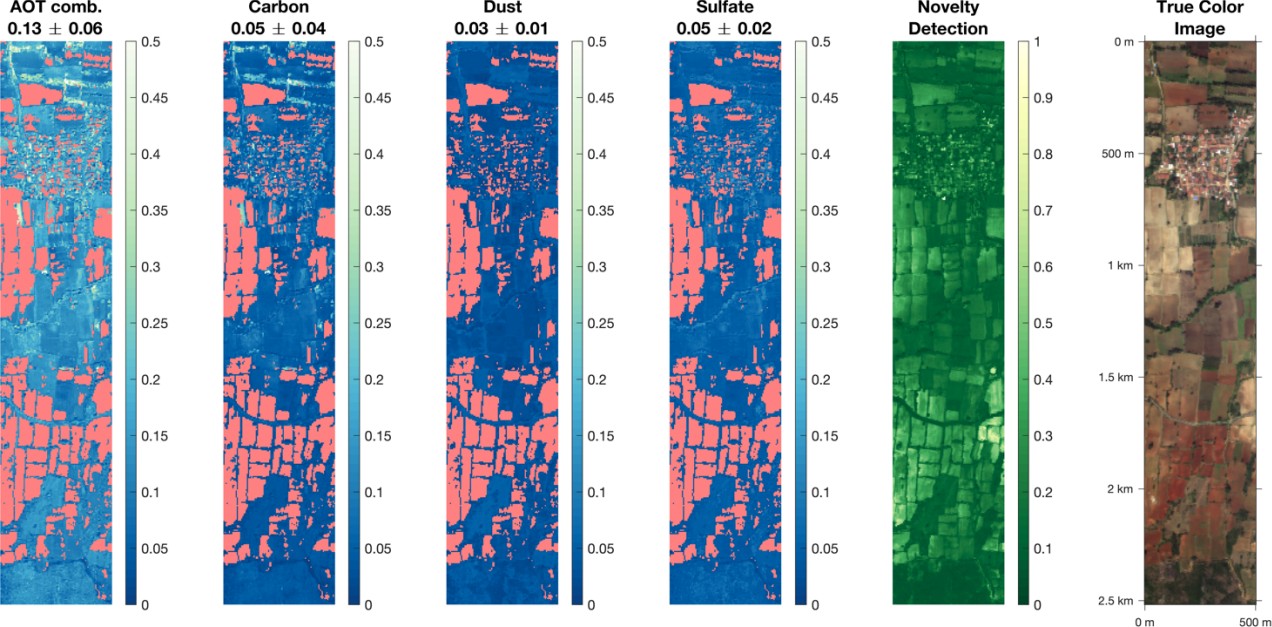

Figure 12: Aerosol retrieval with the model from AVIRIS-NG imagery near Gundlupet, India, 01/10/2016. The median and standard deviation of the retrieval is indicated at the top of each panel. The normalized output of the neural network for novelty detection is shown in the panel, second from the right. Values above a chosen threshold are discarded from the aerosol retrievals and highlighted in red. A true color image of the scene is shown as well for reference.

## 5.4 Comparison to AERONET and MODIS

We compare the combined aerosol retrievals from AVIRIS-NG to AERONET and MODIS retrievals. AERONET is a network of ground-based sun photometers distributed around the globe (Holben et al., 1998). AERONET instruments derive AOT at multiple wavelengths with an uncertainty of 0.01 to 0.02 (Eck et al., 1999). These low uncertainties make AERONET stations a common source for validation of air- and space-borne AOT retrieval (Bilal et al., 2014; Chu et al., 2003; Levy et al., 2013). However, there are sparse AERONET locations in India. We, therefore, add a second source of AOT retrievals to the comparison from MODIS observations. MODIS makes daily and nearly global observations from two platforms, Aqua and Terra. MODIS has a spectral range from 410 nm to 14.5 µm over 36 discrete wavelength bands. Its ground resolution is better than 1 km, depending on the wavelength band (Salomonson et al., 1989). Two algorithms are utilized to derive AOT form MODIS observations. The Dark Target (Kaufman et al., 1997) algorithm is used for dark ground targets such as vegetation and water. The Deep Blue (Hsu et al., 2004, 2006) algorithm is applied to measurements over dark and bright surfaces although it was originally developed for the aerosol retrieval over bright desert regions. Over land, MODIS retrieved AOT has an expected standard error of 0.05 + 15% of AOT (Levy et al., 2013). MODIS has larger uncertainties than AERONET, but the retrievals are in closer spatial and temporal proximity to the AVIRIS-NG flights.

For the period of the 21 AVIRIS-NG flights only three AERONET stations within India were operational. These are Gandhi College at 25.9°N 84.1°E, Jaipur at 26.9°N 75.8°E and Pune at 18.5°N 73.8°E. We make use of the daily means of their Level 2.0 data product, which is cloud-cleared and manually inspected. The locations of all three stations are shown in Figure 13 together with the location of all 21 AVIRIS-NG flights considered in the study. For a given flight we consider the AOT retrieved from all three AERONET stations within 1 and 2 days of the flight date. The time averaged, retrieved AOT of each AERONET station, $\bar{\tau}_{aer\_i}$, is weighted proportionally to the square of the distance, $d_i$, between station and flight:

$$\bar{\tau}_{aer} = \frac{\sum_{i=1}^{3} \bar{\tau}_{aer\_i} * d_i^{-2}}{\sum_{i=1}^{3} d_i^{-2}} \qquad (12)$$

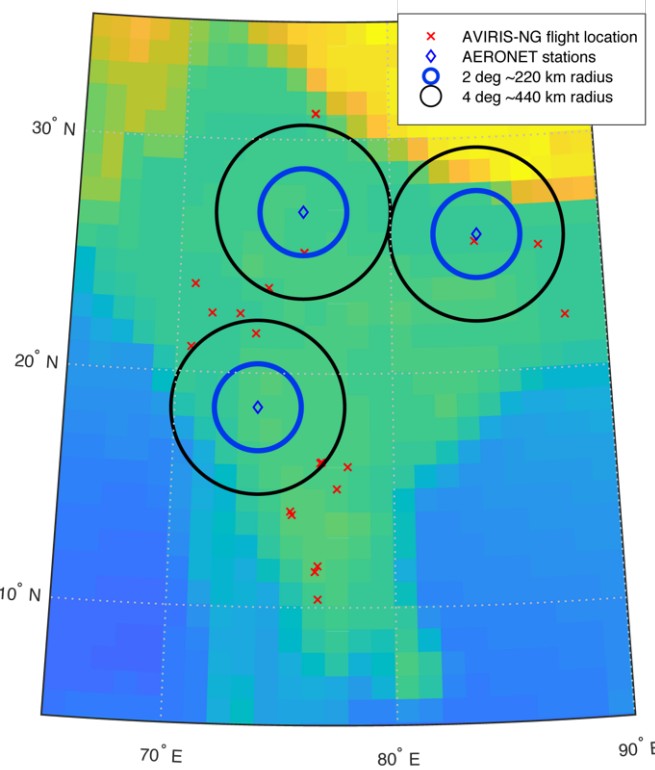

Figure 13: Location of AERONET stations and AVIRIS-NG flights. The AERONET stations are marked with blue diamonds, a 4 deg and 2 deg (approximately 440 km and 220 km) radius around each station is indicated with a black and blue circle. The AVIRIS-NG flight locations are shown with red x. AVIRIS-NG flights outside the circles are not considered for the comparison to AERONET.

The comparison between AOT retrieved by AERONET and the AVIRIS-NG flights is shown in Figure 14 and Figure 15 for AERONET retrievals within 1 day and 2 days, respectively. For the comparison within 1 day of the AVIRIS-NG flights only four AERONET stations reported their measured AOT. Only one comparison falls within the specified 1-day window and is within 2 deg ($\approx$ 220 km) of the flight location (red circle). The three other comparisons are for flights with a distance ranging from 2 deg to 4 deg between the AERONET station and the AVIRIS-NG flight. The standard deviation of all considered AERONET retrievals that we compare to for a given flight is indicated by the vertical bars. The standard deviation within a scene for the analyzed AVIRIS-NG flights is shown with horizontal bars. For the 4 comparisons we find a root mean square difference (RMSD) of 0.09. However, due to the large spatial distance between AERONET stations and the considered AVIRIS-NG flights this value has to be interpreted with caution and comes with large uncertainties. Nevertheless, we included this comparison for completeness and hope to have more collocated flights of AVIRIS-NG and AERONET stations in the future.

Considering AERONET observations within 2 days of the flights we are able to compare eight flights in total with three flights within 2 deg and 5 flights within 2 to 4 deg. The RMSD for all eight comparisons is 0.08. Again, we caution that the distance between AERONET stations and AVIRIS-NG flights is significant. For the comparison within 2 days, the closest comparison has a distance of about 40 km and is shown in Figure 15 (circled red and furthest to the right).

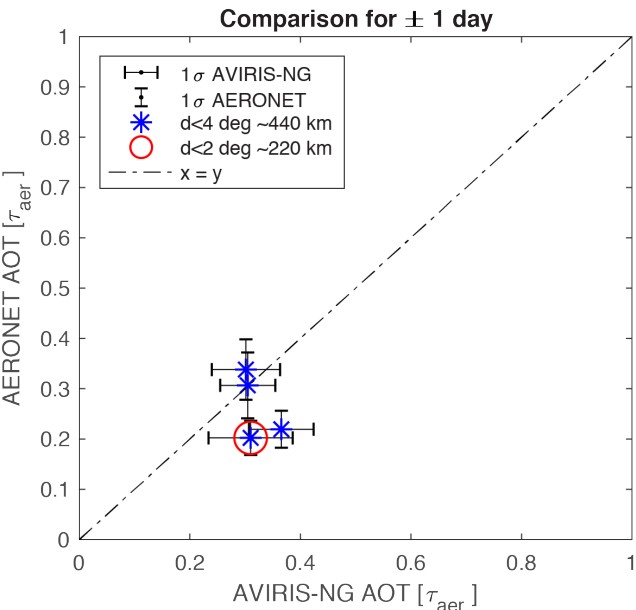

Figure 14: AOT retrieved by AERONET (see Equation 12) and the AOT retrieved from AVIRIS-NG with the model. The standard deviation of the considered AERONET measurements is shown with vertical bars and the standard deviation for the retrieval with AVIRIS-NG with horizontal bars. All comparisons between AERONET and AVIRIS-NG
10  flights are located within 4 deg ($\approx$ 440 km) and within 1 day from each other. The one comparison within 2 deg ($\approx$ 220 km) is circled in red.

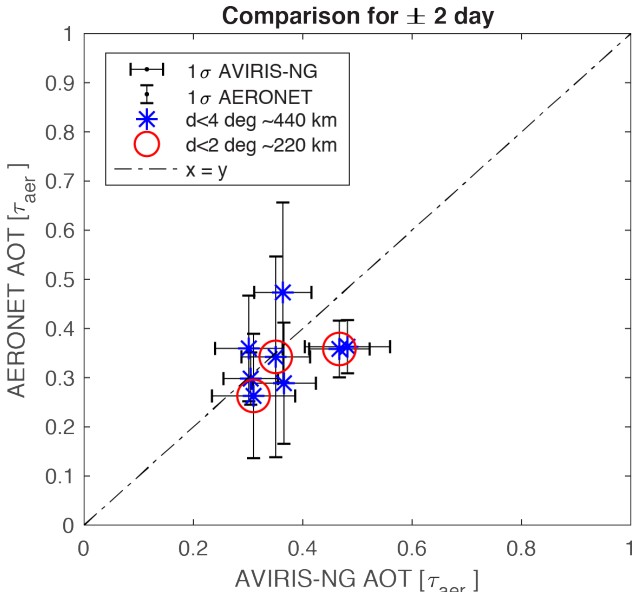

Figure 15: AOT retrieved by AERONET (see Equation 12) and the AOT retrieved from AVIRIS-NG with the model. The standard deviation for AERONET is shown with vertical bars and the standard deviation for the retrieval with horizontal bars. All comparisons between AERONET and AVIRIS-NG flights are located within 4 deg ($\approx$ 440 km) and within 2 day from each other. The three comparisons within 2 deg ($\approx$ 220 km) are circled in red.

For the comparison to MODIS we make use of the Collection 6, 'MODIS/Terra and MODIS/Aqua Level-2 (L2) Aerosol Product' (Levy et al., 2015, 2013). More specifically, we use the science data set 'AOD_550_Dark_Target_Deep_Blue_Combined' within the specified aerosol product. These data have a spatial resolution of 10 x 10 km and are derived utilizing the Dark Target and Deep Blue algorithm. All AOT retrievals come with a Quality Assurance Confidence (QAC), which is a measure of the algorithm performance. The QAC is determined by the number of examined pixel, fitting error and whether the solution falls into realistic physical conditions (Levy et al., 2013). In our study, we only consider derived AOT with the highest QAC = 3 and consider retrievals within 1 day and 0.2 deg $\approx$ 22 km of the AVIRIS-NG flights. The spatiotemporal cutoff is chosen as close in time and space as possible, while avoiding AVIRIS-NG flights with no collocated MODIS retrievals. This results in an average of 55 and minimum of 13 MODIS retrievals per AVIRIS-NG flight that we compare to. The comparison for the 21 AVIRIS-NG flights to the MODIS retrieved AOT is shown in Figure 16. The two AOT retrievals have a significant correlation of 0.81 and a RMSD of 0.12. However, the correlation might be mainly driven by the few high-AOT comparisons. The correlation and RMSD is similar to comparisons between AERONET and MODIS for India, with a correlation of 0.86 and RMSD of 0.19 (Gupta et al., 2018). Furthermore, AVIRIS-NG shows a positive bias of 0.07 compared to MODIS, which itself has a positive bias compared to AERONET (Gupta et al., 2018; Wang et al., 2019). This indicates that the AVIRIS-NG retrievals might overestimate combined AOT. Whether this bias holds true for a larger sample size and whether it is grounded in the model or the calibration of AVIRIS-NG warrants further

investigation. Interestingly, the two outliers at the bottom of Figure 16, where MODIS reports almost no aerosols are only 20 km and 1 day apart from each other. It has to be noted that the presented model was trained purely on radiative transfer calculations and not adjusted or calibrated to match the aerosol retrieval from MODIS or AERONET in any way. As with the comparison to AERONET, the comparison to MODIS comes with caveats. In essence, MODIS faces the same challenges as our model, namely detecting the weak signal of aerosols in the presence of a strong signal from the underlying surface. Furthermore, MODIS AOT retrievals have a different spatial resolution and stem from observations recorded at different times than the AVIRIS-NG flight tracks. Nevertheless, in the absence of higher accuracy collocated measurements we included the comparison to MODIS.

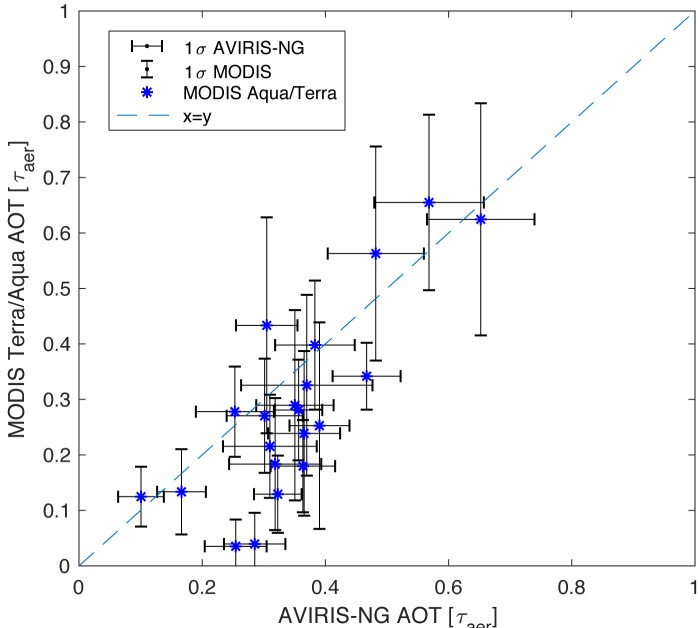

Figure 16: AOT retrieved by MODIS (y-axis) vs AOT retrieved by AVIRIS-NG with the model (x-axis). The standard deviation for MODIS is shown with vertical bars and the standard deviation for the retrieval with horizontal bars.

## 5.5 Comparison to CAMS

We further compare the retrieved AOT to the Copernicus Atmosphere Monitoring Service (CAMS) product. The CAMS system provides global analysis and forecasting of AOT for organic matter, dust and sulfate and is further described in (Benedetti et al., 2009; Morcrette et al., 2009). CAMS accounts for aerosol emissions, transport, sedimentation and deposition of various aerosol types. In contrast to MODIS and AERONET, one can directly compare the CAMS AOT for a specified aerosol type to the retrieved AOT. We make use of the CAMS 'near-real-time' product at a spatial resolution of 0.125° available at: https://apps.ecmwf.int/datasets/data/cams-nrealtime/levtype=sfc/.

Figure 17 shows the comparison for the three considered aerosol types with the CAMS modeled AOT on the y-axis and AVIRIS-NG retrieved AOT on the x-axis. There seems to be general agreement between CAMS and AVIRIS-NG with AVIRIS-NG retrievals being on average 0.03 higher. The standard deviation of the difference between CAMS and AVIRIS-NG for the 21 analyzed scenes is 0.02, 0.04, 0.05 for carbon, dust and sulfate, respectively. For AOT below 0.1, CAMS and AVIRIS-NG differ significantly for carbon and dust with AVIRIS-NG retrieving higher AOT.

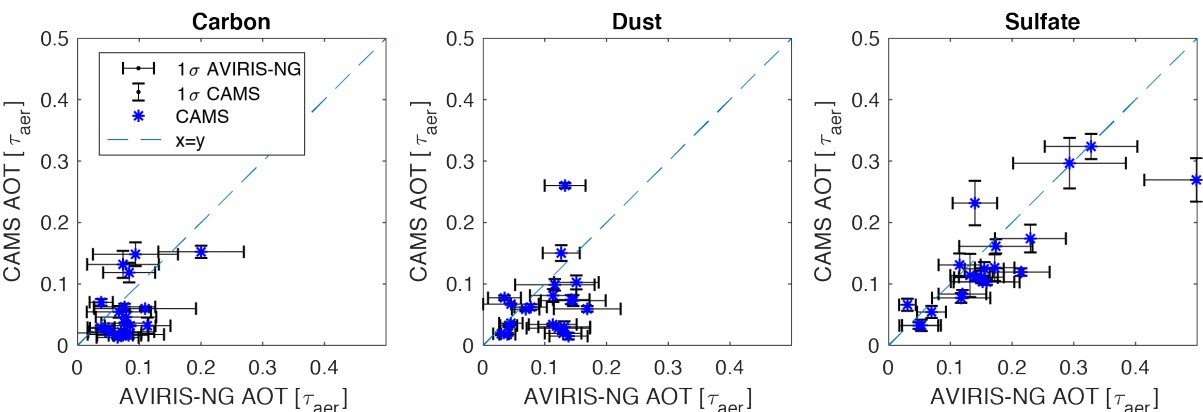

Figure 17: AOT modeled by CAMS (y-axis) vs AOT retrieved from AVIRIS-NG spectra with the neural network (x-axis). The standard deviation of the CAMS modeled AOT within 6 hours and 0.125° of the AVIRIS-NG observations are shown with vertical bars and the standard deviation for the AVIRIS-NG retrievals with horizontal bars.

## 6 Conclusion

We demonstrated the retrieval of AOT from externally mixed dust, sulfate and carbonaceous aerosols from hyperspectral imagery with no a priori information of surface albedo or atmospheric state. We showed how sampling resolution and instrument noise influences the retrieval and, as expected, we find a decrease in model performance for fewer wavelengths and increased instrument noise. These results underline the need for low noise hyperspectral instruments. A sensitivity analysis gave insight in which wavelengths are important and how the neural network compensates for instrument noise; shifting sensitivity to multiple neighboring wavelengths and to longer wavelengths. We applied our model to AVIRIS-NG observations from a recent campaign over India and compared the retrieved AOT to AERONET and MODIS retrievals. The comparison to AERONET show a RMSD in AOT of 0.09 and 0.08 for collocated flights within 1 and 2 days, respectively. The comparison to MODIS finds a RMSD of 0.12. From a test set of radiative transfer calculations, we are able to retrieve AOT independently for dust, sulfate and brown carbon with a standard error of $0.03, 0.03$ and $0.05$, respectively. At execution time the presented neural network methodology can be executed at almost no computational cost. On a high-end consumer laptop (MacBook Pro CPU: i7 at 2.6 GHz) one can extract AOT, with the presented model, at about 250,000 spectra per second.

The results shown here are promising but also underline the difficulties of retrieving aerosol properties, especially over land: aerosol extinction is a weak, slowly varying spectral signal. Hyperspectral measurements can reduce uncertainty in aerosol remote sensing, and we demonstrate that neural networks provide an efficient means for extracting information from large, multi-dimensional data sets, such as hyperspectral data cubes. As future satellite capabilities increase to acquire high spatial resolution hyperspectral data, there is a need to be able to process the large amount of data in a reasonable amount of time. Neural Networks can provide a solution for this task.

**6.1 Future work**

The current set of AVIRIS-NG flights in India has only a limited number of AERONET stations in close proximity to the various flight paths. To further validate our model, more collocated comparisons to AERONET observations are necessary. Deployed on a global platform, such as the upcoming CLARREO pathfinder or HyspIRI mission, many collocated observations with AERONET could systematically validate the retrieval and further improve the model performance through fine tuning. Furthermore, in situ microphysical measurements are necessary to validate the retrieved aerosol types. Finally, the presented methodology can be expanded in the future to retrieve other atmospheric and surface properties, such as water vapor, cloud properties and surface reflectance.

**Code availability**
The code is available at: https://github.com/SteffenMauceri/Aerosol-Retrieval-from-Hyperspectral-observations

**Author contribution**
Mauceri designed, implemented, analyzed and documented the research; Massie provided aerosol parametrization and optical properties; Massie, Kindel and Pilewskie provided guidance for research set-up, analysis and writing.

**Competing interests**
The authors declare that they have no conflict of interest.

**Acknowledgment**
This research was supported in part by NASA Award 80NSSC17K0569

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
