# Peer review of "Neural Network for Aerosol Retrieval from Hyperspectral Imagery"

_Atmospheric Measurement Techniques, 2019_

## Referee Comment (RC1) · Anonymous Referee #1 · 5 Jul 2019

This paper describes the design of a neural network algorithm uses imaging spectrometer measurements to separately retrieve aerosol optical depth (AOD) for three aerosol types (dust, sulphate, brown carbon). The neural network is trained using synthetic spectra and applied to two images from the airborne instrument AVIRIS-NG over India. Overall, the methodology is clearly explained and the paper fits the scope of AMT, but in my opinion the study has a number of shortcomings which should be properly addressed before the paper can be published. The main problem is that, while the NN results are satisfactory on synthetic data, the retrievals on real measurements display large surface features. Other shortcomings are the lack of validation for the per type AOD retrievals and the assumption of spherical dust particles made in the creation of the training set, which may be inaccurate. Below are my detailed comments.

[Figure]

MAIN COMMENTS

- The surface features in the AOD retrieval are really striking, and it would be important to investigate whether the problem may be mitigated by changing the design of the NN. For example, I have the impression that, if you attempt a simple classification of the original image (e.g., into vegetation, bare soil, urban, water classes) and correlate the AOD retrieved on each pixel to the class the pixel belongs to, you may see a strong correlation. If this is the case, then it may mean that you need to pass the result of this simple classification as an additional input to the AOD-retrieval NN, or to train multiple NNs, one per class. This may reduce this effect in your retrievals, which in my opinion is too large at the moment.

- The use of a spherical model for dust aerosols in the generation of the training set may also lead to inaccurate results when the NN is applied to real data (Kalashnikova and Sokolik, 2004, Dubovik et al., 2006, Lee et al., 2017). For this reason, I would recommend to retrain the NN by using a nonspherical model. References:

Kalashnikova, O., and Sokolik, I. N., "Modeling the radiative properties of nonspherical soil-derived mineral aerosols", J. Quant. Spectrosc. Rad. Transfer, 87, 137-166, doi: 10.1016/j.jqsrt.2003.12.026

Dubovik, O. et al. (2006), "Application of spheroid models to account for aerosol particle nonsphericity in remote sensing of desert dust", J. Geophys. Res., 111, D11208, doi: 10.1029/2005JD006619

Lee, J. et al. (2017), "AERONET-based nonspherical dust optical models and effects on the VIIRS Deep Blue/SOAR over water aerosol product", J. Geophys. Res., 122, 10384-10401, doi: 10.1002/2017JD027258

- An additional problem I see is that you try to estimate the AOT for each aerosol type, but you do not provide any indication of the credibility of the per-type retrievals performed with real measurements. While direct observations of "typed" AODs are
probably not available, it would be important to at least have a look of what are the outputs of some reanalysis models (e.g. MERRA-2, CAMS) around the locations you considered and in the same dates. I imagine that these models may have a much coarser spatial grid than your images, but I guess they would be your only possible source of verification.

MINOR COMMENTS

- P1, L21. "... absorption ... are ..." -> "... is ...".

- P1, L24-25. "Instead, a common practice ...". Maybe you mean that one takes the darkest pixel in the image, assumes that the observed radiance over that pixel only comes from the atmosphere and subtracts that radiance from all the other pixels in the image. If so, make that explicit in the paper. If not, clarify what you mean instead.

- P1, L28, "great" -> "larger".

- P2, L24. Please also mention the advances made possible by multiangle polarimetry, which provides an enhanced capability of separating the aerosol signal from the surface signal, and a better sensitivity to the aerosol microphysical parameters (Kokhanovsky et al., 2015, Dubovik et al., 2019). References:

Kokhanovsky, A.A. et al. (2015), "Space-based remote sensing of atmospheric aerosols: The multi-angle spectro-polarimetric frontier", Earth Sci. Rev., 145, 85-116, doi: 10.1016/j.earscirev.2015.01.012

Dubovik, O. et al. (2019), "Polarimetric remote sensing of atmospheric aerosols: Instruments, methodologies, results, and perspectives", J. Quant. Spectrosc. Rad. Transfer, 224, 474-511, doi: 10.1016/j.jqsrt.2018.11.024

- P2, L28. Do you also foresee applying the proposed method to existing satellite imagers such as EO-1 Hyperion, or the recently launched PRISMA?

- P6, Fig.2. I would suggest to change "t_aer" to "tau_aer" in the legend.

- P6. The problem with this study of the sensitivity of the TOA radiance to the aerosol type is that the microphysical properties of the aerosol types are prescribed. Thus, your capability of distinguishing them in your simulation may be greatly overestimated compared to what happens in nature, where I don't think you will see dust, brown carbon etc. always with the same size distribution. Even the refractive index of certain aerosol types (dust in particular) is highly variable, so it would be better to incorporate this variability in the training set (as you already tried to do with the surface properties) in order to have a better hope of making your NN scheme more robust. Note that the aerosol size, in particular, mainly influences the spectral slope of your radiance. Thus, it may well be that your retrieval just tries to distinguish between three "size classes" of aerosols, which you map to "aerosol types" through a rather arbitrary 1:1 correspondence. Also for this reason, it is really important to compare your retrieved aerosol speciation on real data to the outputs of some reanalyses. This would be the only way to obtain at least a preliminary indication that your AOD retrieval distinguished into types actually works in reality.

- P7, L26. I guess you mean "biophysical properties of vegetation".

- P7, L26. Given that your application concerns aerosols, you should also mention previous work on NNs for aerosol retrievals (Radosavljevic et al., 2010, Chimot et al., 2017, Di Noia et al., 2017). References:

Radosavljevic, V. et al. (2010), "A data-mining technique for aerosol retrieval across multiple accuracy measures", IEEE Geosci. Remote Sens. Lett., 7, 411-415, doi: 10.1109/LGRS.2009.2037720

Chimot, J. et al. (2017), "An exploratory study on the aerosol height retrieval from OMI measurements of the 477 nm O2-O2 spectral band using a neural network approach", Atmos. Meas. Tech. 10, 783-809, doi: 10.5194/amt-10-783-2017

Di Noia, A. et al. (2017), "Combined neural network/Phillips-Tikhonov approach to aerosol retrievals over land from the NASA Research Scanning Polarimeter", Atmos.

- P8, L8-9, "training-set" -> "training set", "validation-set" -> "validation set".

- P8, L16. Why are you using radiance and not reflectance as an input? Why are you using ground distance and ground elevation (which should have a relatively minor effect on the top-of-atmosphere radiance or reflectance), but are not using viewing zenith angle and viewing azimuth angle, which may have a greater effect?

- P9, Eqs. 8 and 9. I have the impression that the "n" in Eq. 9 is not the same "n" as in Eq. 8. Please adopt an unambiguous notation and explain the meaning of any symbol you use.

- P9, L6. Add that theta is a vector containing all the weights of the NN (right?). Furthermore, in the next sentence I don't think theta should be the subscript "i" in the L2 norm.

- P9, L7, add "or weight decay" after "L2 regularization".

- P9, L8. "The L2 regularization is weighted" -> "The L2 regularization term R(theta) is weighted"

- P11. Consider splitting Figure 4 into three plots (one per aerosol type). The plot for carbonaceous aerosol looks completely hidden.

- P13, L11-12. In addition to just reducing the number of sampling points, it would be more interesting to also change the spectral resolution of the instrument (I mean, the width of a slit function you may convolve your synthetic spectra with). This would make your setup more similar to that of existing satellite imagers, which typically have a spectral resolution of ~10 nm.

- P14, L7. Since you are using synthetic data with a spherical dust model, it would be important to repeat your experiment with a more realistic model for dust. Otherwise the numbers you provide for the retrieval accuracy are not really meaningful, as they

cannot be really taken as an indication of what would happen in a real scenario.

- L15. You say, "It is inherently difficult to interpret the inner workings of neural networks". Actually, the derivative of the NN output with respect to its input can be computed analytically (Blackwell, 2012, Di Noia et al., 2013). This may enable more systematic sensitivity analyses, as it means that the NN retrieval can be rigorously linearized around its actual input (spectrum + viewing geometry). It may be also useful to feed the values retrieved by the NN back to a radiative transfer model. Combined with the NN input Jacobian mentioned above, this may enable estimating the sensitivity of the NN retrieval to the true state vector (Jiménez and Eriksson, 2001). References:

Blackwell, W. (2012), "Neural network Jacobian analysis for high-resolution profiling of the atmosphere", EURASIP J. Adv. Sig. Proc., 2012, 1-11, doi: 10.1186/1687-6180-2012-71

Di Noia, A. et al. (2013), "Global tropospheric ozone column retrievals from OMI data by means of neural networks", Atmos. Meas. Tech., 6, 895-915, doi: 10.5194/amt-6-895-2013

Jiménez, C. and Eriksson, P. (2001), "A neural network technique for inversion of atmospheric observations from microwave limb sounders", Radio Sci., 36, 941-953, doi: 10.1029/2000RS002561

- P17, L9. "To apply the model to real imagery, one would ideally train the model further on real observations". I don't think this is necessarily true. If your forward simulations and your knowledge of the instrument are realistic enough, using synthetic data should be feasible (again, you can look at Chimot et al. (2017) or Di Noia et al. (2017) for examples). Furthermore, training on real data is guaranteed to introduce sampling biases and co-location errors that may counterbalance the advantage of implicitly incorporating the real instrument characteristics in the training set. Furthermore, for the particular task of retrieving AOD separated into types it may be even impossible to find a training dataset with real observations.

- P18, Section 5.1. There is one aspect that is not totally clear to me. You perform the PCA of AVIRIS-NG observations and retain 16 principal components. Do I correctly understand that what you then pass to the NN are not directly the principal components but the radiance spectra reconstructed from the 16 principal components? If so, please add a sentence somewhere in the section to make this clear.

Apart from this, I have a more fundamental question. You use the PCA as a tool to denoise AVIRIS-NG imagery, which is fine to do, and derive the PCA coefficients from the AVIRIS imagery itself. However, you trained your NN with synthetic spectra, and in order to apply your NN to real observations it is important to make sure that your real data look as similar as possible to the data you used to train the NN. How confident are you that your PCA-based denoising does not change the statistical distribution of the reconstructed radiances compared to that assumed in the training set? It would be interesting to check what happens to the synthetic spectra you used to train the NN if you compress them and reconstruct them with the PCA transformation you derived from the AVIRIS imagery. If they change significantly, then this may be a warning flag that there may be problems when you apply your NN to real measurements pre-processed with your PCA transformation.

- P18, L17. Are you sure components 17 and 18 in Fig. 9 do not contain useful information? They seem to display some "structured" spatial patterns.

- P25, L17. The correlation value looks misleading, as it looks mainly driven by the two high-AOT data points in the upper right.

---

## Referee Comment (RC2) · Anonymous Referee #2 · 5 Aug 2019

Review AMT Manuscript 2019-228

Summary:

The authors present a neural network based aerosol optical thickness retrieval from a hyperspectral imager – with the intent of improving atmospheric correction of land surface reflectances. The network obtains separate estimate of dust, brown carbon, and sulfate AOT as a mixture for each observed spectra. They apply this network to airborne AVIRIS-NG imagery and demonstrate its usage for a variety of aerosol conditions and compare their results those of AERONET. The efficiency of neural networks, combined with the large scale of the remote sensing datasets produced by hyperspectral imagers makes studies such as this one important.

[Figure]

General Comments:

Overall I think this paper is good, but I do think there is relatively little discussion as to why the network performs so poorly for low aerosol optical thicknesses (as indicated in the MODIS comparison). The other comparisons could have appeared better because the collocated datasets simply did not have any occurrences of aerosol optical thicknesses below 0.3. In contrast, the MODIS dataset has numerous occurrences of lower aerosol optical thicknesses. You should look into what might be causing the network to behave this way for low optical thicknesses. It is important to note that MODIS has been shown to have fair agreement with AERONET for low optical thickness conditions in other comparisons (see citations below).

Gupta, P. and Remer, L. A. and Levy, R. C. and Mattoo, S., Validation of MODIS 3 km land aerosol optical depth from NASA's EOS Terra and Aqua missions. 2018, Atmospheric Measurement Techniques, 11, 5, 3145-3159, DOI: 10.5194/amt-11-3145-2018

Yuan Wang, Qiangqiang Yuan, Tongwen Li, Huanfeng Shen, Li Zheng, Liangpei Zhang, Evaluation and comparison of MODIS Collection 6.1 aerosol optical depth against AERONET over regions in China with multifarious underlying surfaces. 2019. Atmospheric Environment, 200, 280-301, https://doi.org/10.1016/j.atmosenv.2018.12.023.

Specific Feedback:

1. The selection of variables for spherical albedo, transmittance and path radiance in equations 1-4 does not seem to match a convention that I am familiar with. It also seems unconventional to refer to radiances with an abbreviation within an equation rather than a consistent variable. The current format makes the equations opaque, when in reality rad_x, t, f, and F are all radiances. 2. Your dust optical property database seems to be from a very old citation. I would recommend using a non-spherical dust model as they have very different scattering properties from spherical scattering. 3. I'm not sure I understand the neural network output structure, are you
retrieving a carbon, dust, and sulfate mixture for all aerosol cases? This should be made more clear, as in some cases/regions this mixture approach may not be appropriate. 4. You refer to the preprocessing in section 3.2 as normalization in a couple of places. This is incorrect, it is typically referred to as "standardization". It is worthwhile to explain why this is useful to perform on neural network inputs before you explain the mathematics. To that end, perhaps it would be helpful to say that the purpose of preprocessing standardization before providing input to a neural network is that it results in a fair comparison of the variability of observations that come from disparate distributions (magnitude and variance). 5. In figures that feature analysis of the validation dataset you need to indicate such in the caption. I think this would apply to Figures 4, 5, 6, 7, and 8. I find that those who are unfamiliar with neural networks and their applications often have difficulty distinguishing between validation dataset figures and those tested on real data unless you very explicitly state that. 6. The novelty detection network is a very clever implementation. I think it would perhaps be useful to further discuss how this works. For example, in many of your images it specifically seems to flag only for very dark surfaces in the true color image – is there an explanation for this behavior? 7. In section 5.4 when you are discussing the comparison to MODIS combined aerosol product you mention that MODIS "uses fewer wavelengths to make this retrieval." I think this may be misleading in a sense. Both of the MODIS aerosol products included in that dataset have a significantly different relationship spectral information and the number of spectral bands used/required than your approach does.

Text Feedback:

1. Page 3 line 20: There is an extra "s" in "TRANSsmittance" in the MODTRAN name. It should be "Transmittance" 2. Page 6 line 8: Missing article in the sentence. "AOT of 1.0 was selected for each aerosol type." should read as "An AOT of 1.0 was selected for each aerosol type." 3. Page 9 line 16: normalization should be standardization. 4. Page 18 line 18: normalized should be standardized.

Figure Feedback:

1. Figure 2: the formatting of optical thickness in the legend is confusing. The variable should either be a tau or AOT. 2. Figure 8: Within each aerosol type the y-axis limits should be consistent. Otherwise it is very difficult to understand how the impact of noise influences the analysis for each of these aerosol types.

———————————————————

---

## Author Comment (AC1) · 16 Sep 2019

Dear Anonymous Referee 1,

Thank you for your time and effort in helping us to improve this paper. We appreciate your comments and suggestions and modified the manuscript accordingly. The responses to all of your comments and marked up manuscript can be found in the supplement.

With kind regards,
Steffen Mauceri and co-authors

**Anonymous Referee #1**

This paper describes the design of a neural network algorithm uses imaging spectrometer measurements to separately retrieve aerosol optical depth (AOD) for three aerosol types (dust, sulphate, brown carbon). The neural network is trained using synthetic spectra and applied to two images from the airborne instrument AVIRIS-NG over India. Overall, the methodology is clearly explained and the paper fits the scope of AMT, but in my opinion the study has a number of shortcomings which should be properly addressed before the paper can be published. The main problem is that, while the NN results are satisfactory on synthetic data, the retrievals on real measurements display large surface features. Other shortcomings are the lack of validation for the per type AOD retrievals and the assumption of spherical dust particles made in the creation of the training set, which may be inaccurate. Below are my detailed comments.

MAIN COMMENTS

- The surface features in the AOD retrieval are really striking, and it would be important to investigate whether the problem may be mitigated by changing the design of the NN. For example, I have the impression that, if you attempt a simple classification of the original image (e.g., into vegetation, bare soil, urban, water classes) and correlate the AOD retrieved on each pixel to the class the pixel belongs to, you may see a strong correlation. If this is the case, then it may mean that you need to pass the result of this simple classification as an additional input to the AOD-retrieval NN, or to train multiple NNs, one per class. This may reduce this effect in your retrievals, which in my opinion is too large at the moment.

Thank you for this suggestion. We investigated whether an explicit classification of the underlying surface type would help the aerosol retrieval. We used a simple 2-layer neural network that classified the surface type and added the classified surface type to the neural network input for the aerosol retrieval. While the surface type classification showed promising results there was no improvement for the aerosol retrieval form the radiative transfer calculations or AVIRIS-NG observations. Our guess is that the NN was always performing an implicit surface type retrieval to separate the aerosol signal from the surface contribution. Thus, the added features were redundant.

However, your later comment: "Why are you using radiance and not reflectance as an input?" made us go back and test whether this could make our neural network more robust when applying it to AVIRIS-NG observations. While the use of reflectance does not improve the aerosol retrieval from our radiative transfer calculations (test set) it mitigates the surface features in the retrieval from AVIRIS-NG. Thus, it is more robust and we now scale the radiances by solar zenith angle and sun-earth distance, similar to deriving apparent reflectance.

Finally, while we agree that it would be nice to completely remove the surface features, less complex atmospheric retrievals from hyperspectral imagery have the same problem. Thus, some remaining surface features in the AOT retrieval are to be expected. We added that to the manuscript:

P21, L6: "Some residual surface features are not entirely unexpected as less challenging atmospheric retrievals from imaging spectroscopy, for example water vapor (Thompson et al., 2015), often contain surface reflectance artifacts."

- The use of a spherical model for dust aerosols in the generation of the training set may also lead to inaccurate results when the NN is applied to real data (Kalashnikova and Sokolik, 2004, Dubovik et al., 2006, Lee et al., 2017). For this reason, I would recommend to retrain the NN by using a nonspherical model. References:
Kalashnikova, O., and Sokolik, I. N., "Modeling the radiative properties of nonspherical soil-derived mineral aerosols", J. Quant. Spectrosc. Rad. Transfer, 87, 137-166, doi: 10.1016/j.jqsrt.2003.12.026
Dubovik, O. et al. (2006), "Application of spheroid models to account for aerosol par- ticle nonsphericity in remote sensing of desert dust", J. Geophys. Res., 111, D11208, doi: 10.1029/2005JD006619
Lee, J. et al. (2017), "AERONET-based nonspherical dust optical models and effects on the VIIRS Deep Blue/SOAR over water aerosol product", J. Geophys. Res., 122, 10384-10401, doi: 10.1002/2017JD027258

We updated our calculation for the dust aerosols to account for their non-sphericity, repeated the radiative transfer calculations, retrained the NN and updated the results of our analysis throughout the manuscript. We did not find systematic differences in the retrieved AOT of dust compared to the original (spherical) assumption.

P6, L4: Content added: "For dust we had to account for its non-spherical shape. We applied the T-matrix code of Mishchenko and Travis, (1998), for randomly oriented particles, to generate the MODTRAN SAP files. The range of ratios of semi-major to semi-minor axes, or aspect ratios (AR), was varied between 1.01 and 1.8. This range contains the representative AR of 1.4 (Okada et al., 2001), while the aspect ratio of 1.01 corresponds to a nearly spherical particle. In our application of the T-matrix code the second mode parameters (i.e. $Rad_2 = 0.83$ μm, $\sigma_2 = 1.84$, see Table 1) were used to specify the size distribution, and the AFCRL 1987 Sand indices are utilized."

- An additional problem I see is that you try to estimate the AOT for each aerosol type, but you do not provide any indication of the credibility of the per-type retrievals performed with real measurements. While direct observations of "typed" AODs are probably not available, it would be important to at least have a look of what are the outputs of some reanalysis models (e.g. MERRA-2, CAMS) around the locations you considered and in the same dates. I imagine that these models may have a much coarser spatial grid than your images, but I guess they would be your only possible source of verification.

That is an excellent idea. We added a comparison to the CAMS model for the three individual aerosol types:

P27, L13: Content added: "**5.5 Comparison to CAMS**
We further compare the retrieved AOT to the Copernicus Atmosphere Monitoring Service (CAMS) product. The CAMS system provides global analysis and forecasting of AOT for organic matter, dust and sulfate and is further described in (Benedetti et al., 2009; Morcrette et al., 2009). CAMS accounts for aerosol emissions, transport, sedimentation and deposition of various aerosol types. In contrast to MODIS and AERONET, one can directly compare the CAMS AOT for a specified aerosol type to the retrieved AOT. We make use of the CAMS 'near-real-time' product at a spatial resolution of 0.125° available at: https://apps.ecmwf.int/datasets/data/cams-nrealtime/levtype=sfc/.

Figure 17 shows the comparison for the three considered aerosol types with the CAMS modeled AOT on the y-axis and AVIRIS-NG retrieved AOT on the x-axis. There seems to be general agreement between CAMS and AVIRIS-NG with AVIRIS-NG retrievals being on average 0.03 higher. The standard deviation of the difference between CAMS and AVIRIS-NG for the 21 analyzed scenes is 0.02, 0.04, 0.05 for carbon, dust and sulfate, respectively. For AOT below 0.1, CAMS and AVIRIS-NG differ significantly for carbon and dust with AVIRIS-NG retrieving higher AOT.

[Figure]

Figure 17: AOT modeled by CAMS (y-axis) vs AOT retrieved by AVIRIS-NG (x-axis). The standard deviation of the CAMS modeled AOT within 6 hours and 0.125° of the AVIRIS-NG observations are shown with vertical bars and the standard deviation for the AVIRIS-NG retrievals with horizontal bars.

MINOR COMMENTS

- P1, L21. "... absorption ... are ..." -> "... is ...".

P1, L21: Changed

- P1, L24-25. "Instead, a common practice . . .". Maybe you mean that one takes the darkest pixel in the image, assumes that the observed radiance over that pixel only comes from the atmosphere and subtracts that radiance from all the other pixels in the image. If so, make that explicit in the paper. If not, clarify what you mean instead.

Removed statement for clarity

P1, L24: Changed to: "Instead, aerosol properties are approximated from visibility (e.g., Gao, Heidebrecht and Goetz, 1993; Adler-Golden et al., 1999) or derived from climatology."

- P1, L28, "great" -> "larger".

P1, L28: Changed

- P2, L24. Please also mention the advances made possible by multiangle polarimetry, which provides an enhanced capability of separating the aerosol signal from the surface signal, and a better sensitivity to the aerosol microphysical parameters (Kokhanovsky et al., 2015, Dubovik et al., 2019). References:
Kokhanovsky, A.A. et al. (2015), "Space-based remote sensing of atmospheric aerosols: The multi-angle spectro-polarimetric frontier", Earth Sci. Rev., 145, 85-116, doi: 10.1016/j.earscirev.2015.01.012
Dubovik, O. et al. (2019), "Polarimetric remote sensing of atmospheric aerosols: Instruments, methodologies, results, and perspectives", J. Quant. Spectrosc. Rad. Transfer, 224, 474-511, doi: 10.1016/j.jqsrt.2018.11.024

P2, L23: Added information: "Other approaches aim to use the vast information content from space-borne multiangle polarimetric observations that provide enhanced capability of separating aerosol signal from surface signal, and a better sensitivity to aerosol microphysical parameters. However, retrieving aerosol properties from such observations is highly complex and operational products have not yet reached the accuracy implied by theoretical calculations (Dubovik et al., 2019; Kokhanovsky et al., 2015)."

- P2, L28. Do you also foresee applying the proposed method to existing satellite imagers such as EO-1 Hyperion, or the recently launched PRISMA?

P2, L29: Added both instruments: "To increase accuracy of global aerosol retrievals, we propose a retrieval algorithm that will be applicable to current and future hyperspectral space-borne instruments, such as Hyperspectral Precursor and Application Mission (PRISMA) (Labate et al., 2009), EO-1 Hyperion (Folkman et al., 2001), Climate Absolute Radiance and Refractive Observatory (CLARREO) (Wielicki et al., 2013) and the Hyperspectral Infrared Imager (HyspIRI) (Lee et al., 2015)."

- P6, Fig.2. I would suggest to change "t_aer" to "tau_aer" in the legend.

P6, Fig.2: Changed "t_aer" to "$\tau$". Thanks for catching that.

- P6. The problem with this study of the sensitivity of the TOA radiance to the aerosol type is that the microphysical properties of the aerosol types are prescribed. Thus, your capability of distinguishing them in your simulation may be greatly overestimated compared to what happens in nature, where I don't think you will see dust, brown carbon etc. always with the same size distribution. Even the refractive index of certain aerosol types (dust in particular) is highly variable, so it would be better to incorporate this variability in the training set (as you already tried to do with the surface properties) in order to have a better hope of making your NN scheme more robust. Note that the aerosol size, in particular, mainly influences the spectral slope of your radiance. Thus, it may well be that your retrieval just tries to distinguish between three "size classes" of aerosols, which you map to "aerosol types" through a rather arbitrary 1:1 correspondence. Also for this reason, it is really important to compare your retrieved aerosol speciation on real data to the outputs of some reanalyses. This would be the only way to obtain at least a preliminary indication that your AOD retrieval distinguished into types actually works in reality.

We agree that radiative transfer calculations will always under sample the variety found in nature. However, to limit the number of radiative transfer calculations and computation time we limited calculations to three aerosol types with a representative size distribution for every aerosol type. Exploring different refractive indices and size distributions is beyond the scope of this paper. However, as you suggested, we compared our output to the CAMS reanalysis model, which shows promising results. (see Maine Comments)

- P7, L26. I guess you mean "biophysical properties of vegetation".

P7, L13: Changed to: "biophysical properties of vegetation"

- P7, L26. Given that your application concerns aerosols, you should also mention previous work on NNs for aerosol retrievals (Radosavljevic et al., 2010, Chimot et al., 2017, Di Noia et al., 2017). References:
Radosavljevic, V. et al. (2010), "A data-mining technique for aerosol retrieval across multiple accuracy measures", IEEE Geosci. Remote Sens. Lett., 7, 411-415, doi: 10.1109/LGRS.2009.2037720
Chimot, J. et al. (2017), "An exploratory study on the aerosol height retrieval from OMI measurements of the 477 nm O2-O2 spectral band using a neural network approach", Atmos. Meas. Tech. 10, 783-809, doi: 10.5194/amt-10-783-2017
Di Noia, A. et al. (2017), "Combined neural network/Phillips-Tikhonov approach to aerosol retrievals over land from the NASA Research Scanning Polarimeter", Atmos. Meas. Tech., 10, 4235-4252, doi: 10.5194/amt-10-4235-2017

P8, L13: Information added: "Neural networks have also been applied to retrieve aerosol layer height from Ozone Monitoring Instrument (OMI) observations (Chimot et al., 2017), estimate multiple aerosol parameters as a prior for an iterative Phillips-Tikhonov retrieval (Di Noia et al., 2017) and to estimate AOT from MODIS observations (Lary et al., 2009; Radosavljevic et al., 2010)."

- P8, L8-9, "training-set" -> "training set", "validation-set" -> "validation set".

Changed throughout the manuscript

- P8, L16. Why are you using radiance and not reflectance as an input? Why are you using ground distance and ground elevation (which should have a relatively minor effect on the top-of-atmosphere radiance or reflectance), but are not using viewing zenith angle and viewing azimuth angle, which may have a greater effect?

We switched from standardizing radiance by its mean to scaling radiance by sun-earth distance and SZA. This makes the retrieval more robust to the AVIRIS-NG observations and is similar to using reflectance as our input. We use ground elevation and surface-sensor-distance since they provide additional information to the NN. While both variables might have a limited impact on top-of-atmosphere radiances, our radiative transfer calculations are performed for the altitude of the airborne AVIRIS-NG instrument where the effect of ground distance and elevation is greater. We don't provide the viewing zenith angle and viewing azimuth angle as an input to the NN since we assume a nadir looking sensor.

-P9, Eqs. 8 and 9. I have the impression that the "n" in Eq. 9 is not the same "n" as in Eq. 8. Please adopt an unambiguous notation and explain the meaning of any symbol you use.

P9, Eq9: Changed the second "n" into an "m" and added both to the text: "

$$R(\theta) = \|\theta\|_2 = \sqrt{\sum_{i=1}^{m} \theta_i^2} \tag{1}$$

For our network $\hat{Y}_j$ and $Y_j$ are the $n$ true and predicted AOT, respectively. We further add the L2 norm $\|\theta\|_2$ of the vector of the $m$ neural network weights, $\theta$, to our cost function (see Equation 9), also referred to as *L2 regularization* or *weight decay*."

- P9, L6. Add that theta is a vector containing all the weights of the NN (right?). Furthermore, in the next sentence I don't think theta should be the subscript "i" in the L2 norm.

P9, L17: Removed the subscript "i" and added that $\theta$ is a vector: "For our network $\hat{Y}_j$ and $Y_j$ are the $n$ true and predicted AOT, respectively. We further add the L2 norm $\|\theta\|_2$ of the vector of the $m$ neural network weights, $\theta$, to our cost function (see Equation 9), also referred to as *L2 regularization* or *weight decay*."

- P9, L7, add "or weight decay" after "L2 regularization".

P9, L18, Changed

- P9, L8. "The L2 regularization is weighted" -> "The L2 regularization term R(theta) is weighted"

P10, L1: Changed

- P11. Consider splitting Figure 4 into three plots (one per aerosol type). The plot for carbonaceous aerosol looks completely hidden.

P11, Fig 4: Great idea. We split Figure 4 into three plots and changed the visualization to a 'heatmap' for even better interpretability:

[Figure]

 Figure 4: AOT for carbon, dust and sulfate aerosols, retrieved by the model vs true AOT from the test set. The cyan line shows the linear fit to the data with slope and y-intercept given in the respective titles.

- P13, L11-12. In addition to just reducing the number of sampling points, it would be more interesting to also change the spectral resolution of the instrument (I mean, the width of a slit function you may convolve your synthetic spectra with). This would make your setup more similar to that of existing satellite imagers, which typically have a spectral resolution of ~10 nm.

We repeated our test cases for a spectral resolution of 10 nm. To limit the total number of neural networks we had to train we reduced the original exploration of sampling resolution and instrument noise slightly. However, we found no significant differences to the test cases with a spectral resolution of 5 nm.

P13, L18: We added this information to the text: "While we found dependencies of retrieval performance to varying amounts of noise and number of wavelength channels, the spectral resolution had no significant effect. On average the models trained with a spectral resolution of 5 nm had a standard error in retrieved AOT that was only 0.001 smaller than for the cases with a spectral resolution of 10 nm.  Therefore, we limit the following discussion to the results of the 12 neural networks trained on radiative transfer calculations with the AVIRIS-NG spectral resolution of approximately 5 nm and note that these values are also representative for an instrument with a 10 nm spectral resolution."

- P14, L7. Since you are using synthetic data with a spherical dust model, it would be important to repeat your experiment with a more realistic model for dust. Otherwise the numbers you provide for the retrieval accuracy are not really meaningful, as they cannot be really taken as an indication of what would happen in a real scenario.

We updated our calculation for the dust aerosols to account for their non-sphericity (see Maine Comments).

- L15. You say, "It is inherently difficult to interpret the inner workings of neural networks". Actually, the derivative of the NN output with respect to its input can be computed analytically (Blackwell, 2012, Di Noia et al., 2013). This may enable more systematic sensitivity analyses, as it means that the NN retrieval can be rigorously linearized around its actual input (spectrum + viewing geometry). It may be also useful to feed the values retrieved by the NN back to a radiative transfer model. Combined with the NN input Jacobian mentioned above, this may enable estimating the sensitivity of the NN retrieval to the true state vector (Jiménez and Eriksson, 2001). References:
Blackwell, W. (2012), "Neural network Jacobian analysis for high-resolution profiling of the atmosphere", EURASIP J. Adv. Sig. Proc., 2012, 1-11, doi: 10.1186/1687-6180- 2012-71
Di Noia, A. et al. (2013), "Global tropospheric ozone column retrievals from OMI data by means of neural networks", Atmos. Meas. Tech., 6, 895-915, doi: 10.5194/amt-6- 895-2013
Jiménez, C. and Eriksson, P. (2001), "A neural network technique for inversion of atmospheric observations from microwave limb sounders", Radio Sci., 36, 941-953, doi: 10.1029/2000RS002561

Our approach is very similar to the proposed Jacobian analysis by Blackwell. However, the Jacobian analysis (and our approach) are limited in that they only allow to calculate sensitivities for a certain operating point (input that we linearize around). Since the neural network is non-linear it is "inherently difficult" to understand how the neural network will behave in any situation (compared to e.g. linear regression). Blackwell acknowledges the importance of choosing the right operating point in Chapter 6 and 7. Nevertheless, we agree that neural networks are partially interpretable and tried to shed some light on the proposed neural network with our preformed sensitivity analysis.

P15, L9: Citation added for Blackwell, 2012.

- P17, L9. "To apply the model to real imagery, one would ideally train the model further on real observations". I don't think this is necessarily true. If your forward simulations and your knowledge of the instrument are realistic enough, using synthetic data should be feasible (again, you can look at Chimot et al. (2017) or Di Noia et al. (2017) for examples). Furthermore, training on real data is guaranteed to introduce sampling biases and co-location errors that may counterbalance the advantage of implicitly incorporating the real instrument characteristics in the training set. Furthermore, for the particular task of retrieving AOD separated into types it may be even impossible to find a training dataset with real observations.

The introduction of a sampling bias is something we hadn't considered when making the above statement. We removed the paragraph about fine tuning.

- P18, Section 5.1. There is one aspect that is not totally clear to me. You perform the PCA of AVIRIS-NG observations and retain 16 principal components. Do I correctly understand that what you then pass to the NN are not directly the principal components but the radiance spectra reconstructed from the 16 principal components? If so, please add a sentence somewhere in the section to make this clear.

P18, L18: Thanks for catching that. Clarified paragraph and changed to: "The first 16 principal components explain approximately 99.9% of the variability in the observations. We reconstruct the AVIRIS-NG observed radiances from these principal components. That effectively removes principal components higher than 16 from all analyzed AVIRIS-NG imagery. Afterwards, the radiance for every pixel is treated as an independent observation and scaled and standardized (Equation 10 and 11) to match the training set.

Apart from this, I have a more fundamental question. You use the PCA as a tool to denoise AVIRIS-NG imagery, which is fine to do, and derive the PCA coefficients from the AVIRIS imagery itself. However, you trained your NN with synthetic spectra, and in order to apply your NN to real observations it is important to make sure that your real data look as similar as possible to the data you used to train the NN. How confident are you that your PCA-based denoising does not change the statistical distribution of the reconstructed radiances compared to that assumed in the training set? It would be interesting to check what happens to the synthetic spectra you used to train the NN if you compress them and reconstruct them with the PCA transformation you derived from the AVIRIS imagery. If they change significantly, then this may be a warning flag that there may be problems when you apply your NN to real measurements pre- processed with your PCA transformation.

We investigated the sensitivity of the neural network to removing principal components higher than 16 and found no evidence of systematic differences. The denoising was motivated to eliminate vertical stripes in the aerosol retrievals that initially showed up in the scenes of Figure 11 and 12. (P18, L26) "Experiments with more and fewer principal components indicated that the model was insensitive to the exact number of remaining principal components."

- P18, L17. Are you sure components 17 and 18 in Fig. 9 do not contain useful information? They seem to display some "structured" spatial patterns.

The first 16 principal components capture 99.9% of the variability. We acknowledged in the manuscript that the choice for the cut-off is rather arbitrary: (P18, L23) "We acknowledge that the choice of retaining the first 16 principal components is rather arbitrary and should ideally be made on a per flight basis. However, for practical reasons we decided to use one threshold for all imagery considered in this study. The threshold is a tradeoff between removing valuable information and reducing noise. Experiments with more and fewer principal components indicated that the model was insensitive to the exact number of remaining principal components."

- P25, L17. The correlation value looks misleading, as it looks mainly driven by the two high-AOT data points in the upper right.

P26, L18: We added the following disclaimer: "However, the correlation might be mainly driven by the few high-AOT comparisons."

Furthermore, we tested the correlation for significance and found the correlation to be significant at the 0.05 p-value.

[revised manuscript text omitted]

---

## Author Comment (AC2) · 16 Sep 2019

Dear Anonymous Referee 2,

Thank you for your time and effort in helping us to improve this paper. We appreciate your comments and suggestions and modified the manuscript accordingly. The responses to all of your comments and marked up manuscript can be found in the supplement.

With kind regards,
Steffen Mauceri and co-authors

**Anonymous Referee #2**

The authors present a neural network based aerosol optical thickness retrieval from a hyperspectral imager – with the intent of improving atmospheric correction of land surface reflectances. The network obtains separate estimate of dust, brown carbon, and sulfate AOT as a mixture for each observed spectra. They apply this network to airborne AVIRIS-NG imagery and demonstrate its usage for a variety of aerosol conditions and compare their results those of AERONET. The efficiency of neural networks, combined with the large scale of the remote sensing datasets produced by hyperspectral imagers makes studies such as this one important.

General Comments

Overall I think this paper is good, but I do think there is relatively little discussion as to why the network performs so poorly for low aerosol optical thicknesses (as indicated in the MODIS comparison). The other comparisons could have appeared better because the collocated datasets simply did not have any occurrences of aerosol optical thicknesses below 0.3. In contrast, the MODIS dataset has numerous occurrences of lower aerosol optical thicknesses. You should look into what might be causing the network to behave this way for low optical thicknesses. It is important to note that MODIS has been shown to have fair agreement with AERONET for low optical thickness conditions in other comparisons (see citations below).

Gupta, P. and Remer, L. A. and Levy, R. C. and Mattoo, S., Validation of MODIS 3 km land aerosol optical depth from NASA's EOS Terra and Aqua missions. 2018, Atmospheric Measurement Techniques, 11, 5, 3145-3159, DOI: 10.5194/amt-11-3145- 2018

Yuan Wang, Qiangqiang Yuan, Tongwen Li, Huanfeng Shen, Li Zheng, Liangpei Zhang, Evaluation and comparison of MODIS Collection 6.1 aerosol optical depth against AERONET over regions in China with multifarious underlying surfaces. 2019. Atmo- spheric Environment, 200, 280-301, https://doi.org/10.1016/j.atmosenv.2018.12.023

As a result of comments by Referee 1 we updated our approach to using reflectance rather than normalized radiance. This resolved some of the differences between MODIS and AVIRIS-NG. However, there are still differences between both retrievals. We added a comparison to CAMS (as suggested by Referee 1) that shows that the bias comes from overestimating carbon and dust for low AOT (see P28, Fig 17). We added more discussion but can only speculate were the bias comes from. The MODIS retrievals have a different spatial resolution and are recorded at a different time than the AVIRIS-NG flights. Both of these make a direct comparison challenging. To investigate whether the differences between AVIRIS-NG and MODIS are systematic will necessitate more collocated flights of AVIRIS-NG with AEORNET stations on future missions.

P26, L20: Discussion Added: "The two AOT retrievals have a significant correlation of 0.81 and a RMSD of 0.12. However, the correlation might be mainly driven by the few high-AOT comparisons. The correlation and RMSD is similar to comparisons between AERONET and MODIS for India, with a correlation of 0.86 and RMSD of 0.19 (Gupta et al., 2018). Furthermore, AVIRIS-NG shows a positive bias of 0.07 compared to MODIS, which itself has a positive bias compared to AERONET (Gupta et al., 2018; Wang et al., 2019). This indicates that the AVIRIS-NG retrievals might overestimate combined AOT. Whether this bias holds true for a larger sample size and whether it is grounded in the model or the calibration of AVIRIS-NG warrants further

investigation. Interestingly, the two outliers at the bottom of Figure 16, where MODIS reports almost no aerosols are only 20 km and 1 day apart from each other."

P27, L5: Possible reason for differences added: "Furthermore, MODIS AOT retrievals have a different spatial resolution and stem from observations recorded at different times than the AVIRIS-NG flight tracks"

Specific Feedback:

1. The selection of variables for spherical albedo, transmittance and path radiance in equations 1-4 does not seem to match a convention that I am familiar with. It also seems unconventional to refer to radiances with an abbreviation within an equation rather than a consistent variable. The current format makes the equations opaque, when in reality rad_x, t, f, and F are all radiances.

P4: We updated the notation and changed:
- radiance: $Rad\_x$ to $L\_x$
- two-way transmittance: $t$ to $\tau$
- spherical albedo: $\sigma$ to $\rho$
- path radiance: $f$ to $L_P$
- at sensor radiance: $F$ to $L$

2. Your dust optical property database seems to be from a very old citation. I would recommend using a non-spherical dust model as they have very different scattering properties from spherical scattering.

We updated our calculations for the dust aerosols to account for their non-sphericity, repeated the radiative transfer calculations, retrained the NN and updated the results of our analysis throughout the manuscript. We did not find systematic differences in the retrieved AOT of dust compared to the original (spherical) assumption. Nevertheless, the model is now trained with non-spherical dust.

Content added: "To account for the fact that dust particles can be aspherical, we applied the T-matrix code of Mishchenko (Mishchenko and Travis, 1998), for randomly oriented particles, to generate MODTRAN SAP files for a range of ratios of semi-major to semi-minor axes, or aspect ratios (AR), between 1.01 and 1.8. This range contains the representative AR of 1.4 (Okada et al., 2001), while the aspect ratio of 1.01 corresponds to a nearly spherical particle. In our application of the T-matrix code the second mode parameters (i.e. $Rad_2 = 0.83$ µm, $\sigma_2 = 1.84$, see Table 1) were used to specify the size distribution, and the AFCRL 1987 Sand indices are utilized."

3. I'm not sure I understand the neural network output structure, are you retrieving a carbon, dust, and sulfate mixture for all aerosol cases? This should be made more clear, as in some cases/regions this mixture approach may not be appropriate.

Our neural network retrieves carbon, dust and sulfate aerosols independently for all cases. This can be interpreted as an external mixture of those aerosols. If only one aerosol type is present in a given scene the AOT for the other aerosol types will be zero. We agree that this would not be appropriate for some cases or regions.

Added "independently retrieved" to sentence that describes the neural network output for clarification:

P9, L2: The output of the network is the **independently retrieved** AOT of the three aerosol types.

4. You refer to the preprocessing in section 3.2 as normalization in a couple of places. This is incorrect, it is typically referred to as "standardization". It is worthwhile to explain why this is useful to perform on neural network inputs before you explain the mathematics. To that end, perhaps it would be helpful to say that the purpose of preprocessing standardization before providing input to a neural network is that it results in a fair comparison of the variability of observations that come from disparate distributions (magnitude and variance).

We reorganized the description and replaced "normalization" with "standardization":

P10, L9: "We then scale the radiance of a particular observation, $L_{obs\_j}$, by dividing through the cosine of the SZA and multiplying with the square of the Sun-Earth distance, $d$, (Equation 10). This scales the magnitude of $L_{obs\_j}$ while preserving its spectral shape. Afterwards, we standardize the scaled observations, $R_j$, for the training process. During training, this results in a better conditioned cost function and allows the neural network to converge faster to a solution. Standardizing is performed by subtracting the mean, $\mu_\lambda$, and dividing by the standard deviation, $\sigma_\lambda$, at every wavelength, (Equation 11). The mean and standard deviation was calculated from the complete set of scaled radiative transfer calculations.

$$R_j = \frac{L_{obs\_j} * d_j^2}{\cos(SZA)} \tag{1}$$

$$\overline{R_j} = \frac{R_j - \mu_\lambda}{\sigma_\lambda} \quad with \quad \mu_\lambda = mean(R)_\lambda \quad and \quad \sigma_\lambda = std(R)_\lambda \tag{2}$$

5. In figures that feature analysis of the validation dataset you need to indicate such in the caption. I think this would apply to Figures 4, 5, 6, 7, and 8. I find that those who are unfamiliar with neural networks and their applications often have difficulty distinguishing between validation dataset figures and those tested on real data unless you very explicitly state that.

Updated captions to state that the data comes from the test set

6. The novelty detection network is a very clever implementation. I think it would perhaps be useful to further discuss how this works. For example, in many of your images it specifically seems to flag only for very dark surfaces in the true color image – is there an explanation for this behavior?

The dark surfaces that stick out are water. This is not surprising since the neural network was trained only on land surface reflectances. While the magnitude is much lower the network for novelty detection also identified some bright soil areas in the scene from 01/10/2016

P20, L14: Added an example to help the reader understand how the neural network for novelty detection works: "During training, the neural network learns to minimize this error. For example, the neural network learns that the radiance at 2100 nm is highly correlated with the radiance at 2300 nm. Thus, it can reconstruct (decompress) both radiances with only one value passed in from the Bottleneck layer with little error. Once the neural network is trained and applied to previously unseen features it will compress and decompress features that are similar to the training set (high correlation between 2100 nm and 2300 nm) with a smaller error than features that are different (low correlation between 2100 nm and 2300 nm)."

P20, L3: Added citation for further reading: "Japkowicz, N., Myers, C., Gluck, M. and others: A novelty detection approach to classification, in IJCAI, vol. 1, pp. 518–523., 1995."

P21, L13: Added explanation for the identified water features: "The detection of water by the neural network for novelty detection is to be expected, since the reflectance of water is very different to most land surfaces and was not part of the training set."

7. In section 5.4 when you are discussing the comparison to MODIS combined aerosol product you mention that MODIS "uses fewer wavelengths to make this retrieval." I think this may be misleading in a sense. Both of the MODIS aerosol products included in that dataset have a significantly different relationship spectral information and the number of spectral bands used/required than your approach does.

We agree that this might be misleading and removed it.

Text Feedback:

1. Page 3 line 20: There is an extra "s" in "TRANSsmittance" in the MODTRAN name. It should be "Transmittance"

P3, L24: Thanks for catching that. Removed "s"

2. Page 6 line 8: Missing article in the sentence. "AOT of 1.0 was selected for each aerosol type." should read as "An AOT of 1.0 was selected for each aerosol type."

P6, L19: Added "An"

3. Page 9 line 16: normalization should be standardization.

P10, L11: Changed to "standardized"

4. Page 18 line 18: normalized should be standardized.

P18, L23: Changed to "standardized"

Figure Feedback:

1. Figure 2: the formatting of optical thickness in the legend is confusing. The variable should either be a tau or AOT.

P7, Fig.2: Changed "t_aer" to "$\tau$"

2. Figure 8: Within each aerosol type the y-axis limits should be consistent. Otherwise it is very difficult to understand how the impact of noise influences the analysis for each of these aerosol types.

P17, Fig.8: Changed all y-axis limits to be the same

[revised manuscript text omitted]

---

## Author Response (AR2)

Dear Andrew,

Thank you for your comments and the comments of the anonymous reviewer. The paper improved a lot since its initial submission. We addressed all comments in red below. One co-author was not sure whether it is ok to write "deg" instead of " ° ". If that is an issue, I can update that across the manuscript and figures.
Let me know if you have any remaining questions.

Thanks
Steffen and Co-authors

**Comments from Anonymous Reviewer:**
- P3, L11. Also mention in the introduction that you compared your typed AOT retrievals to CAMS.
P3, L11. Added CAMS to description of Section 5: "In Section 5 we apply the inverse model to AVIRIS-NG observations and compare results to AERONET and MODIS retrieved AOT **and the CAMS analysis product**."

- P8, L8. After "multilayer perceptrons", cite: Werbos, P. J.: Beyond Regression: New Tools for Prediction and Analysis in the Behavioral Sciences, PhD thesis, Harvard University, Cambridge, MA, United States, 1974
P8, L8. Citation added

- P8, L17. I would replace "operations" with "functional elements".
P8, L17. Changed to: "A multilayer perceptron is comprised of many individual **functional elements**, or *neurons*, that …"

- P10, L10. Aren't you actually multiplying by the square ratio between the actual Sun-Earth distance d and the mean Sun-Earth distance d0? Otherwise your modified radiance would be expressed in a strange unit (a radiance times the square of a length). Note that, even if d0=1 AU and thus numerically d/d0 equals d in AU, dimensionally what you are taking is still the ratio between the two quantities. [Editor's note: I think the reviewer is correct here, please check.]
P10, L10. Thanks for catching that! Changed to: "We then scale the radiance of a particular observation, $L_j$, by dividing through the cosine of the $SZA_j$ and multiplying with the square of the **ratio between** Sun-Earth distance, $d_j$**, and mean Sun-Earth distance, $d_0$,** (Equation 10)."

Eq 10 updated: "$R_j = \dfrac{L_j * \left(\frac{d_j}{d_0}\right)^2}{\cos(SZA_j)}$ "

- P11, L5.Maybe you can emphasize that, if you look at Fig. 2, the spectral signature of sulphate and brown carbon look very similar (except for a scaling factor - which means that it may be possible to confuse sulfate with a certain AOT with brown carbon with a higher AOT) whereas dust has a markedly different spectrum thanks to its absorption at shorter wavelengths. Therefore, one may expect some degree of confusion between sulfate and carbon and a better capability of identifying dust.

P11, L6. Interpretation added: "Thus, the model has the smallest uncertainties for the retrieval of dust and sulfate, which are less strongly absorbing across most of the relevant spectral range, compared to carbon. Distinguishing the weaker signal from carbon from the surface contribution might be more challenging for the neural network and lead to the overall larger uncertainties for carbon. Additionally, the spectral signature of carbon and sulfate show high similarities at shorter wavelengths, except for a scaling factor (see Figure 2). This might further challenge the neural network to distinguishing the two. For wavelengths ranging from 500 nm to 600 nm, absorption of dust increases with decreasing wavelength, while the opposite is true for carbon and sulfate. This unique spectral signature of dust, compared to the other two aerosol types, might be another reason for the smaller standard error of its retrieval."

- Section 5.5. The sulfate AOT plot looks quite encouraging. It would be also interesting to see in how many of the 21 considered scenes CAMS and your NN agree on which is the aerosol type with the largest AOT.

P28, L7. Comparison added: "We further evaluate the agreement between AVIRIS-NG and CAMS for which aerosol type has the largest, second largest and smallest AOT in a given scene. For 16 out of the 21 compared scenes AVIRIS-NG and CAMS agree on which aerosol type dominates with the largest AOT (see Figure 18). 14 out of 21 compared scenes agree on which aerosol type has the second largest AOT and 16 out of 21 agree on which aerosol type has the smallest AOT. Thus, if one were to use the proposed neural network to classify the dominant aerosol type from AVIRIS-NG observations, the neural network would have an estimated accuracy of 76%.

[Figure]

Figure 18: Number of scenes where CAMS and AVIRIS-NG agree and disagree for which aerosol type has the largest (1st), 2nd largest and 3rd largest AOT. The total number of analyzed scenes is 21. Thus, the 16 in the bottom left corresponds to 16 scenes in which AVIRIS-NG and CAMS identified the same aerosol type as having the largest AOT, out of 21 scenes."

**Comments from Andrew Sayer:**
Page 2 line 8: the reference should appear as "Pope et al.", not "Pope III et al". The surname is "Pope" and the author is the third of that name in the family. I am not sure if you are using BibTeX for references but if so a guide for typesetting names containing constructs like that is here: https://tex.stackexchange.com/questions/557/how-should-i-type-author-names-in-a-bib-file I think the correct syntax for the author name in the bibliography file is "Pope, III, C. A."

P2, L8: Citation changed to "Pope et al."

Page 2 line 18: it would be better to state that MODIS/MISR etc have "near-global" rather than "global" coverage, as there remain several systematic sampling gaps.
P2, L18: Changed to "near-global"

Equation 4: from this, it would be good to note somewhere in the manuscript that the surface is assumed to be Lambertian. (A familiar reader can infer this from the Equation, but it would be good to state it directly in the text too.)
P3, L31: Added information that surface is assumed to be Lambertian: "The three simulations are then used to calculate at sensor radiance for any given surface albedo, **which is assumed to be Lambertian,** utilizing …"

Equation 10: this quantity is commonly referred to as "top of atmosphere reflectance" in the aerosol remote sensing community (typically after also scaling by pi). I realise you may not wish to use the term reflectance due to the potential confusion with surface reflectance. I leave it up to you whether you wish to note this identity in the manuscript.
P10, L11: Added that the two are similar to each other: "This scales the magnitude of $L_j$ while preserving its spectral shape **and is similar to deriving the *top of atmosphere reflectance***."

Page 11 line 5: the comment about SSA is wavelength-dependent, as dust is fairly strongly absorbing in the blue and UV. So I would add a qualifier about which wavelength range you are talking about here. One might instead right that dust and sulfate "are less strongly absorbing across most of the relevant spectral range" than carbon. Also, I think you mean "uncertainty" rather than "accuracy" here (since we're discussing total error, not bias).
P11, L6: Replaced "accuracy" with "uncertainty" and expanded possible explanation for differences in the standard errors for the three aerosol types: "Thus, the model has the smallest uncertainties for the retrieval of dust and sulfate, which are less strongly absorbing across most of the relevant spectral range, compared to carbon. Distinguishing the weaker signal from carbon from the surface contribution might be more challenging for the neural network and lead to the overall larger uncertainties for carbon. Additionally, the spectral signature of carbon and sulfate show high similarities at shorter wavelengths, except for a scaling factor (see Figure 2). This might further challenge the neural network to distinguishing the two. For wavelengths ranging from 500 nm to 600 nm, absorption of dust increases with decreasing wavelength, while the opposite is true for carbon and sulfate. This unique spectral signature of dust, compared to the other two aerosol types, might be another reason for the smaller standard error of its retrieval."

Figure 13: what is the background image here? I didn't see that stated; it should also have a colour bar if the colours are to be interpreted quantitatively.
P24, L2: Added information to Figure description. We did not add a color bar of this topographic map since it would distract from the information we try to show (spatial distribution of AERONET stations with respect to the AVIRIS-NG flights). "
[revised manuscript text omitted]
_j$  and multiplying with the square of the ratio between Sun-Earth distance, $d_j$, and mean Sun-Earth distance, $d_0$ , (Equation 10). This scales the magnitude of $L_j$ while preserving its spectral shape and is similar to deriving the *top of atmosphere reflectance*. Afterwards, we standardize the scaled observations, $R_j$, for the training process. During training, this results in a better conditioned cost function and allows the neural network to converge faster to a solution. Standardizing is performed by subtracting the mean, $\mu_\lambda$, and dividing by the standard deviation, $\sigma_\lambda$, at every wavelength, (Equation 11). The mean and standard deviation was calculated from the complete set of radiative transfer calculations.

$$R_j = \frac{L_j * \left(\frac{d_j}{d_0}\right)^2 \cancel{d_j^2}}{\cos\left(SZA_j \cancel{SZA}\right)} \tag{10}$$

$$R_j = \frac{R_j - \mu_\lambda}{\sigma_\lambda} \quad with \quad \mu_\lambda = mean(R_j)_\lambda \quad and \quad \sigma_\lambda = std(R_j)_\lambda \tag{11}$$

**3.3 Training, Validation and Test**

The MODTRAN radiance samples were split into a trainings, validation and test set. The validation and test set contain 10,000 randomly chosen samples each and the training set consists of 280,000 samples. Training is performed with Googles' TensorFlow framework (Abadi et al., 2016). We gradually minimize the cost function by adjusting the randomly initialized weights and bias terms with the gradient-based optimizer Adam from Kingma and Ba (2014), at a learning rate of 0.001. During training we evaluate the neural network performance on the validation set and update the model architecture and training parameters. Once, the cost function cannot be further minimized, training is complete.

**4 Results and Discussion**

After training of the neural network is completed, we evaluate its performance on the test set. For the samples in the test set, that were not present during training, we find a linear correlation coefficient of 0.87, 0.98 and 0.96 for the AOT of carbon,

5   dust and sulfate, respectively (see Figure 4). The standard error for carbon-, dust- and sulfate aerosols is 0.05, 0.02 and 0.03, respectively. Thus, the model has the smallest uncertainties  for the retrieval of dust and sulfate, which are less strongly absorbing across most of the relevant spectral range,  compared to carbon. Distinguishing the weaker signal from carbon from the surface contribution might be more challenging for the neural network and lead to the overall larger uncertainties for carbon. Additionally, the spectral signature of carbon and sulfate show high

10   similarities at shorter wavelengths, except for a scaling factor (see Figure 2). This might further challenge the neural network to distinguishing the two. For wavelengths ranging from 500 nm to 600 nm, absorption of dust increases with decreasing wavelength, while the opposite is true for carbon and sulfate. This unique spectral signature of dust, compared to the other two aerosol types, might be another reason for the smaller standard error of its retrieval.

[revised manuscript text omitted]

We further evaluate the agreement between AVIRIS-NG and CAMS for which aerosol type has the largest, second largest and smallest AOT in a given scene. For 16 out of the 21 compared scenes AVIRIS-NG and CAMS agree on which aerosol type dominates with the largest AOT (see Figure 18). 14 out of 21 compared scenes agree on which aerosol type has the second largest AOT and 16 out of 21 agree on which aerosol type has the smallest AOT. Thus, if one were to use the proposed neural network to classify the dominant aerosol type from AVIRIS-NG observations, the neural network would have an estimated accuracy of 76%.

[Figure]

Figure 17: AOT modeled by CAMS (y-axis) vs AOT retrieved from AVIRIS-NG spectra with the neural network (x-axis). The standard deviation of the CAMS modeled AOT within 6 hours and 0.125° of the AVIRIS-NG observations are shown with vertical bars and the standard deviation for the AVIRIS-NG retrievals with horizontal bars.

[Figure]

Commented [SM3]: Figure added

Figure 18: Number of scenes where CAMS and AVIRIS-NG agree and disagree for which aerosol type has the largest (1st), 2nd largest and 3rd largest AOT. The total number of analyzed scenes is 21. Thus, the 16 in the bottom left corresponds to 16 scenes in which AVIRIS-NG and CAMS identified the same aerosol type as having the largest AOT, out of 21 scenes.

[revised manuscript text omitted]

World Health Organisation: Burden of disease from the joint effects of household and ambient Air pollution for 2016, 2018.

World Health Organization: Air quality guidelines: global update 2005, World Health Organization., 2006.

Xiao, Z., Liang, S., Wang, J., Chen, P., Yin, X., Zhang, L. and Song, J.: Use of general regression neural networks for generating the GLASS leaf area index product from time-series MODIS surface reflectance, IEEE Trans. Geosci. Remote Sens., 52(1), 209–223, 2014.